EMBO
*reports*

# The NSP3 protein of SARS-CoV-2 binds fragile X mental retardation proteins to disrupt UBAP2L interactions

Dimitriya H Garvanska[1], R Elias Alvarado [ID] [2,3], Filip Oskar Mundt[1], Richard Lindqvist[4], Josephine Kerzel Duel[1], Fabian Coscia [ID] [1], Emma Nilsson [ID] [4], Kumari Lokugamage[2], Bryan A Johnson[2], Jessica A Plante[2,5], Dorothea R Morris[2,3], Michelle N Vu[2], Leah K Estes[2], Alyssa M McLeland[2], Jordyn Walker[2,5], Patricia A Crocquet-Valdes [ID] [6], Blanca Lopez Mendez [ID] [1], Kenneth S Plante[2,5], David H Walker[6], Melanie Bianca Weisser[1], Anna K Överby [ID] [4], Matthias Mann[1], Vineet D Menachery[2,5] & Jakob Nilsson [ID] [1]✉

## Abstract

**Viruses interact with numerous host factors to facilitate viral replication and to dampen antiviral defense mechanisms. We currently have a limited mechanistic understanding of how SARS-CoV-2 binds host factors and the functional role of these interactions. Here, we uncover a novel interaction between the viral NSP3 protein and the fragile X mental retardation proteins (FMRPs: FMR1, FXR1-2). SARS-CoV-2 NSP3 mutant viruses preventing FMRP binding have attenuated replication in vitro and reduced levels of viral antigen in lungs during the early stages of infection. We show that a unique peptide motif in NSP3 binds directly to the two central KH domains of FMRPs and that this interaction is disrupted by the I304N mutation found in a patient with fragile X syndrome. NSP3 binding to FMRPs disrupts their interaction with the stress granule component UBAP2L through direct competition with a peptide motif in UBAP2L to prevent FMRP incorporation into stress granules. Collectively, our results provide novel insight into how SARS-CoV-2 hijacks host cell proteins and provides molecular insight into the possible underlying molecular defects in fragile X syndrome.**

**Keywords** SARS-CoV-2; Fragile X Syndrome; UBAP2L; Stress Granules; NSP3
**Subject Categories** Microbiology, Virology & Host Pathogen Interaction; Signal Transduction

## Introduction

Viruses encode a limited number of proteins and are thus highly dependent on interactions with cellular host factors to efficiently replicate in host cells (Davey et al, 2011; Simonetti et al, 2023). Furthermore, viruses dampen innate immune signaling by interfering with distinct steps in the cellular signaling cascades. A common target of viruses is to disrupt the formation of stress granules which have been implicated as signaling hubs for antiviral signaling (Guan et al, 2023; Lloyd, 2013; McCormick and Khaperskyy, 2017; Miller, 2011). Stress granules, large membrane-less protein-RNA assemblies, form in the cytoplasm in response to various stress signals, including viral infections (Protter and Parker, 2016). Induced by host translation inhibition, stress granules are known to sequester RNA. However, they are also composed of a large number of RNA-binding proteins including G3BP1/2 and UBAP2L that play a key role in nucleating and coordinating stress granule formation (Cirillo et al, 2020; Jain et al, 2016; Markmiller et al, 2018; Youn et al, 2018). Importantly, over 250 host proteins have been implicated in playing a role during stress granule formation, highlighting the complexity of this process (Markmiller et al, 2018).

Given the link to antiviral defenses, viruses have developed strategies to disrupt stress granule formation and even hijacked these factors to facilitate their replication (Eiermann et al, 2020; Jayabalan et al, 2023). For coronaviruses, viral proteins have been implicated in disrupting stress granule formation, including the nucleocapsid protein (N) and accessory ORFs (Nakagawa et al, 2018; Peng et al, 2008; Rabouw et al, 2016). Indeed, we and others recently showed that the N protein of SARS-CoV-2 and SARS-CoV contains an ΦxFG motif that binds to G3BP1/2 to disrupt stress granule formation (Biswal et al, 2022; Huang et al, 2021;

[1]Novo Nordisk Foundation Center for Protein Research, Faculty of Health and Medical Sciences, University of Copenhagen, Copenhagen, Denmark. [2]Department of Microbiology and Immunology, University of Texas Medical Branch, Galveston, TX, USA. [3]Institute for Translational Sciences, University of Texas Medical Branch, Galveston, TX, USA. [4]Department of Clinical Microbiology, Umeå University, Umeå, Sweden. [5]World Reference Center of Emerging Viruses and Arboviruses, University of Texas Medical Branch, Galveston, TX, USA. [6]Department of Pathology, University of Texas Medical Branch, Galveston, TX, USA. ✉E-mail: jakob.nilsson@cpr.ku.dk

Kruse et al, 2021; Yang et al, 2023). This represents one of several mechanisms SARS-CoV-2 uses to antagonize cellular antiviral mechanisms (Dolliver et al, 2022; Xia et al, 2020). Similarly, members of both the old and new world alphaviruses bind G3BP1/2 proteins through ΦxFG motifs in the hypervariable domains of the NSP3 protein to facilitate viral replication complex assembly and disruption of stress granule formation (Foy et al, 2013; Scholte et al, 2015). Interestingly, eastern equine encephalitis virus (EEEV) binds to both G3BP1/2 and the FMR1/FXR1-2 (FMRPs) through distinct sequences in their hypervariable domain (Frolov et al, 2017; Kim et al, 2016). FMRPs, RNA-binding proteins, are also recruited to stress granules and deregulated expression of or mutations in FMR1 results in fragile X syndrome, the most common form of inherited mental retardation (Colak et al, 2014; De Boulle et al, 1993; Fu et al, 1991; Verkerk et al, 1991). The exact underlying molecular cause of fragile X syndrome is still not fully understood, but mutations in UBAP2L and G3BP1/2 have also been linked to mental retardation, highlighting the importance of stress granules to host functions (Jia et al, 2022).

In this manuscript, we uncover a novel direct interaction between SARS-CoV-2 NSP3 and FMRPs and uncover its role during viral infection in molecular detail.

## Results

### The SARS-CoV-2 NSP3 protein binds to FMRPs

To understand SARS-CoV-2 interactions with cellular host factors, we focused on the multifunctional ~200 kDa NSP3 protein (Lei et al, 2018). NSP3, the largest coronavirus protein, has multiple domains with functions essential for viral infection. While there is significant variation across the coronavirus (CoV) family, eight domains are common to all coronavirus NSP3 proteins including two ubiquitin-like domains, the hypervariable region, macrodomain, papain-like protease, zinc-finger domain, and the two Y domains of unknown function (Fig. 1A). Notably, NSP3 multimers also form pore-like structures in double-membrane vesicles housing the CoV replication complex (Wolff et al, 2020). Given the large size, transmembrane domains, and critical roles during infection, NSP3 has been difficult to study, and its many functions remain poorly understood.

For this reason, we chose to explore SARS-CoV-2 NSP3's interaction with host factors. The pore structure indicates that the N-terminal portion of NSP3 protrudes into the cytoplasm and likely facilitates numerous interactions with host proteins (Wolff et al, 2020). Therefore, we expressed and affinity-purified a YFP-tagged version of the cytosolic part of NSP3 and compared the associated proteins to a control purification using quantitative label-free mass spectrometry. Strikingly, the most prominent cellular host factors co-purifying with NSP3 was the RNA-binding protein FMR1 and the highly related FXR1 and FXR2 proteins (FMR1, FXR1 and FXR2 collectively referred to here as FMRPs) (Fig. 1B; Dataset EV1). The NSP3-FMRP interaction has also been noted in other high throughput screens of SARS-CoV-2 proteins, but its role in viral infection is unknown (Almasy et al, 2021; Kim et al, 2023).

To determine which region of NSP3 is interacting with FMRPs, we immunopurified a panel of YFP-tagged NSP3 fragments expressed in HeLa cells and monitored binding to endogenous

FXR1 by western blotting. Our results revealed that the N-terminal 181 amino acids of NSP3 were capable of binding FXR1 (Fig. EV1A). Due to its large size, multiple domains, and sequence diversity, it is difficult to compare NSP3 similarity across the coronavirus family. To determine if the observed NSP3/FMRP interaction occurs in other CoVs, we generated N-terminal fragments of five human coronaviruses outside the Sarbeco coronaviruses. However, immunopurifications with these N-terminal regions of NSP3 from these other coronaviruses revealed that tight binding of NSP3 to FMRPs was restricted to SARS-CoV-2 (Fig. EV1B).

To more precisely identify the binding site of FMRPs we mutated blocks of ten amino acids to Ala in the intrinsically disordered regions of NSP3 1–181. In total, we generated nine mutants with two mutants covering the intrinsically disordered N-terminus and 7 the intrinsically disordered region following the Ubl1 domain. These constructs where expressed in cells together with the viral N protein and immunopurified. Our results established that binding to FMRPs was mediated by a stretch of 20 amino acids in the hypervariable region (HVR) following the Ubl1 domain (Fig. 1C). Notably, mutation of this region did not affect the NSP3-N interaction in agreement with predictions from the structure of the complex (Bessa et al, 2022). The observation that N binding was not affected also argued against misfolding of NSP3 upon mutation of the unstructured region binding to FMRPs. While there is variation in the HVR, this 20-amino acid sequence was conserved in all of the Sarbeco family of coronaviruses (Fig. EV1C); no similar sequence motifs were found in other coronaviruses. Together, our finding defines the motif in NSP3 binding to FMRPs and indicates its conservation across the Sarbeco virus family.

### SARS-CoV-2 NSP3 mutant viruses have attenuated replication in vitro

Having established an interaction between NSP3 and FMRPs within a 20-amino acid region in the HVR, we next sought to determine the impact of this interaction on SARS-CoV-2 infection. Utilizing our reverse genetic system (Xie et al, 2021b; Xie et al, 2020), we generated two recombinant SARS-CoV-2 mutant viruses (mut-1 and mut-2) with the 10 alanine mutations in NSP3 preventing FMRP binding (C3 and C4 in Fig. 1C). Both NSP3 mutants were recovered with normal stock titers and plaque morphology. Examining VeroE6 cells, the NSP3 mutant viruses were slightly attenuated in replication 24 h post infection (HPI) relative to the WT SARS-CoV-2 (Fig. 1D). While attenuation was mostly absent at 48 HPI, the reduced capacity of the NSP3 mutants in the type I interferon (IFN) deficient VeroE6 was noteworthy. We subsequently examined replication in Calu3 cells, an IFN-responsive respiratory cell line (Fig. 1E). Similar to VeroE6 cells, NSP3 mut-1 and mut-2 were attenuated compared to WT SARS-CoV-2. Together, the data show that disruption of this 20-amino acid stretch in NSP3 attenuated SARS-CoV-2 replication in vitro.

While type I IFN is a major driver of the antiviral response in cells additional antiviral mechanisms exist. The attenuation of the NSP3 mutants in IFN-deficient VeroE6 cells suggests interferon-stimulated genes (ISGs) may not drive attenuation. To evaluate IFN sensitivity, we pretreated VeroE6 cells with type I IFN and infected with SARS-CoV-2 WT and NSP3 mutants (Fig. 1F). Following ISG activation, WT viruses had a 0.5 to 1.25 log reduction in viral titer compared to untreated. However, NSP3 mutants had a two- to

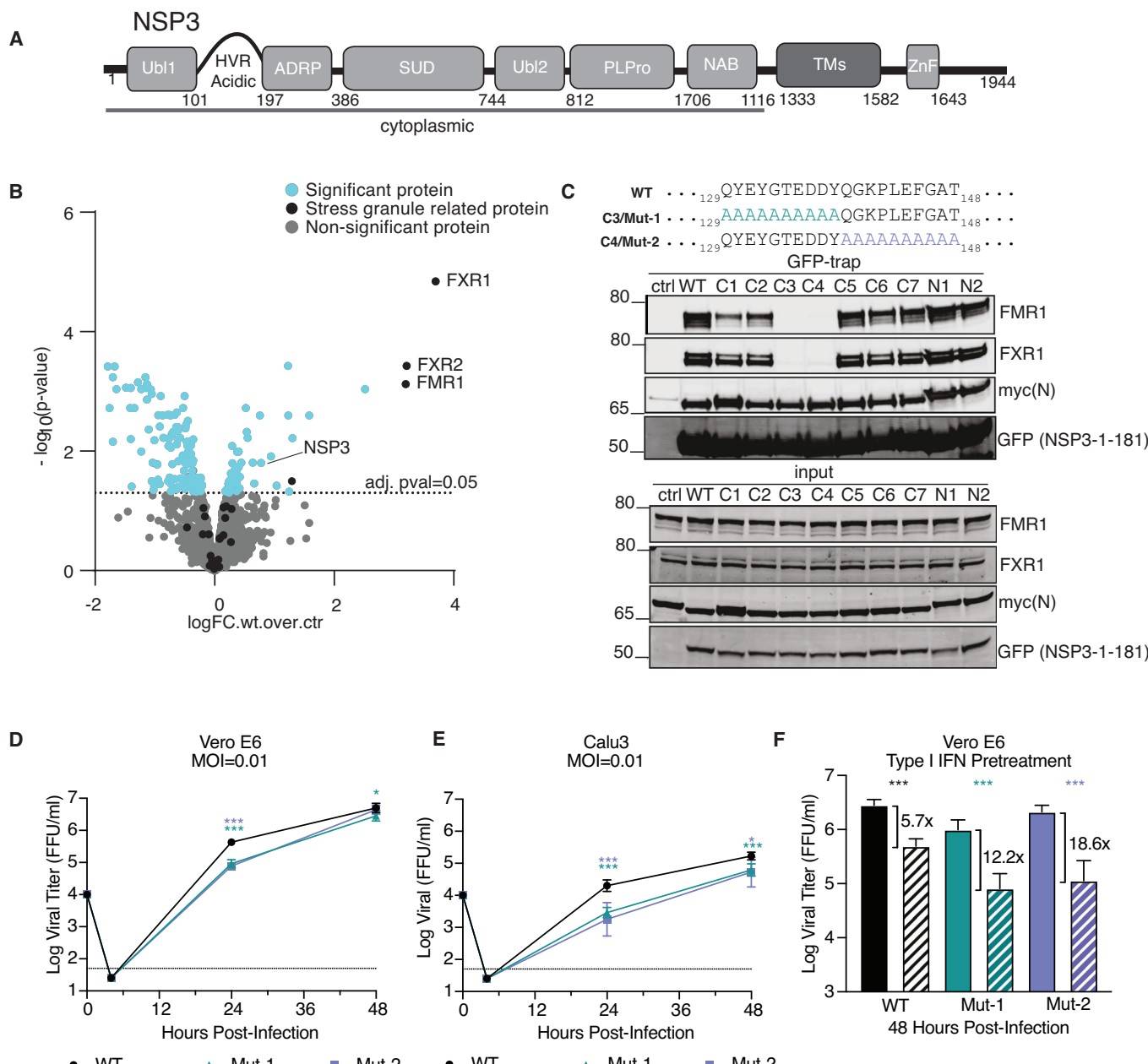

**Figure 1. An NSP3–FMRPs interaction is required for efficient SARS-CoV-2 replication.**

(**A**) Schematic of NSP3 protein with distinct domains indicated. (**B**) Interactome of the cytoplasmic domains of NSP3 in HeLa cells. Data from four technical replicates. (**C**) Interaction of NSP3 mutants with FXR1, FMR1 and myc-tagged N protein to map binding sites. Each variant has ten amino acids mutated to Alanine. Representative of two biological replicates. (**D, E**) VeroE6 cells or Calu3 cells were infected with the indicated SARS-CoV-2 viruses and viral titers measured at 24 and 48 h post infection ($n = 6$ from two experiments each with three biological replicates). (**F**) VeroE6 cells were pretreated with control (solid) or 100 unit of type I IFN (hashed) for 16 h and then infected with the indicated SARS-CoV-2 viruses and viral titers measured after 48 h ($n = 6$ from two experiments each with three biological replicates). The fold change relative to control is shown. Data information: In (**B**), a two-sided unpaired $t$ test was used for statistical analysis. (**D–F**) Statistical analysis measured by two-tailed Student's $t$ test: ****$P < 0.0001$,***$P < 0.001$, *$P < 0.05$. (**D–F**) Data are presented as mean with SD. Source data are available online for this figure.

threefold increase in sensitivity compared to WT. Compared to IFN-sensitive SARS-CoV-2 NSP16 mutants (>10,000-fold titer reduction) (Schindewolf et al, 2023), the modest susceptibility of NSP3 mutants suggests potential antagonism of an antiviral pathway unrelated to the ISG response, governing most of the attenuation.

## NSP3 mutants are attenuated during early in vivo infection

To investigate the role of the NSP3-FMRP interaction in vivo, we next infected 3- to 4-week-old hamsters with WT and NSP3 mutant SARS-CoV-2 evaluating weight loss and disease over a 7 day time

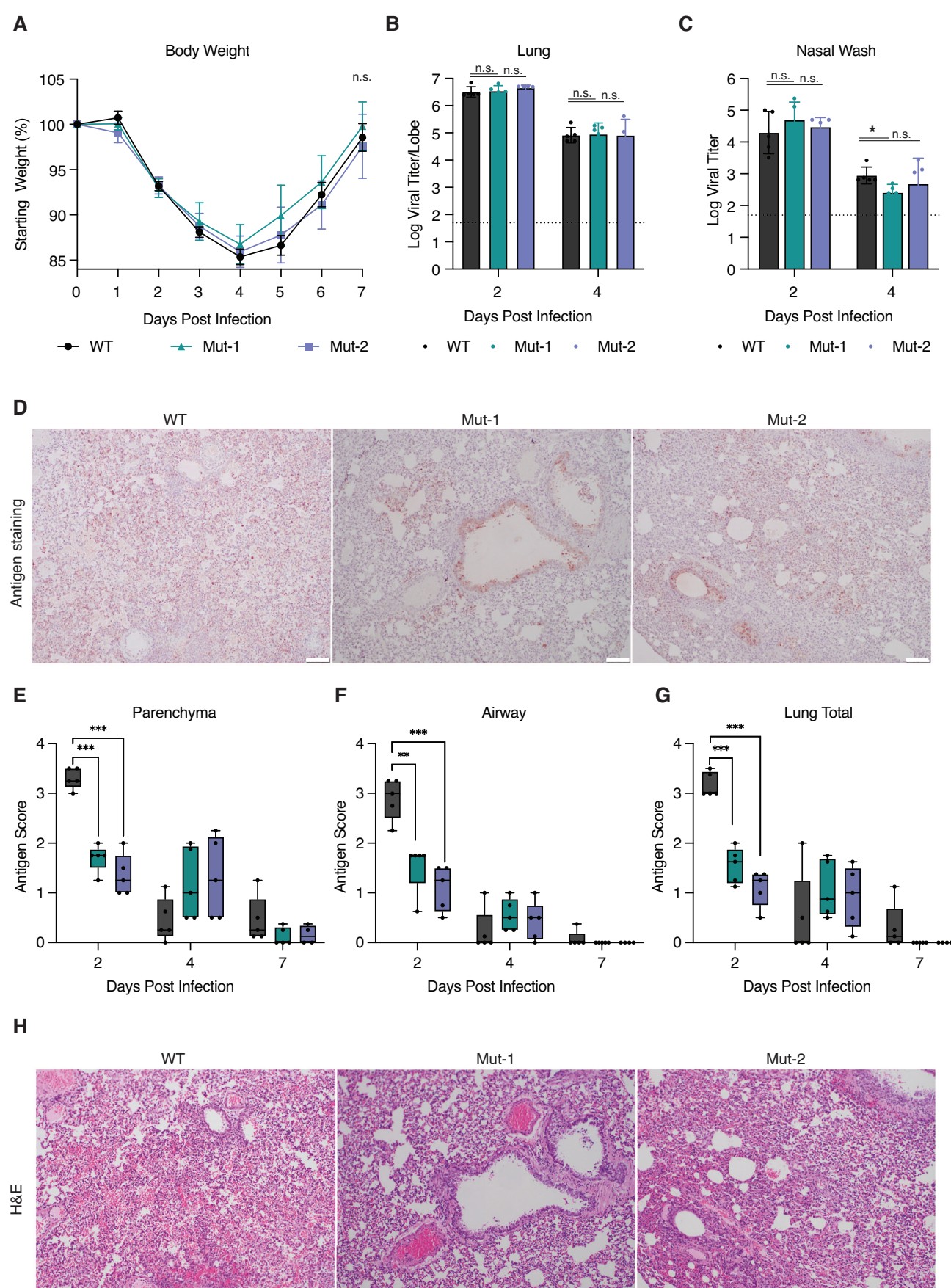

◄    **Figure 2.    In vivo characterization of SARS-CoV-2 virus unable to bind FMRPs.**

(A) Golden Syrian hamsters were infected with $10^5$ plaque-forming units (PFU) of WT SARS-CoV-2 ($n = 15$), NSP3 mutants ($n = 15$), or mock (PBS, $n = 15$) and monitored for weight loss and signs of disease over a 7 day time course. (B, C) At days 2, 4, and 7 post infection, hamsters ($n = 5$ individual hamsters) were nasal washed and subsequently euthanized and tissue collected to assay viral titers from (B) lung or (C) nasal wash. (D) Representative lung tissue sections stained for viral antigen (nucleocapsid) at day 2 from WT, NSP3 Mut-1 and Mut-2 infected animals. See Fig. EV2B for full images. (E–G) Antigen staining was scored on a 4 points scale for the parenchyma, airway, and by total for WT and NSP3 mutants on days However, at day 4, these trends reversed with both NSP3 mutants showing more antigen staining than 2, 4, and 7 in a blinded manner. Each data point is representative of the average score from two lung section from each hamsters in the group ($n = 5$ individual hamsters). (H) Day 2 lung tissue sections parallel to (D) were stained for H&E and demonstrated increased immune cell infiltrate and more severe lesions at day 2 post in WT-infected hamsters compared to NSP3 mutant-infected animals. See Fig. EV2B for full images. Data information: Statistical analysis was measured by two-tailed Student's $t$ test: n.s. not significant, ***$P < 0.001$, **$P < 0.01$, *$P < 0.05$. (A–C) Mean and SEM is shown while in (E–G) Min/Max plotted, center is mean, and all points shown. In (D, H), scale bar is 100 µm. Source data are available online for this figure.

course (Fig. EV2A). In addition, animals were nasal washed, euthanized and tissue collected for further analysis at days 2, 4, and 7. Hamsters infected with either the NSP3 Mut-1 or Mut-2 showed no significant changes in either weight loss or disease relative to WT-infected animals (Fig. 2A). Similarly, we detected no significant changes in viral titers in either the nasal wash or lungs infected with NSP3 mutants as compared to WT (Fig. 2B,C). These results indicate that despite attenuation in vitro, the NSP3 mutations have only a minimal impact on SARS-CoV-2 pathogenesis and viral replication in vivo.

Despite similar viral titers, histopathology results suggest attenuation of the NSP3 mutants early during in vivo infection. Examining antigen staining, WT SARS-CoV-2 infected hamsters show viral antigen throughout the lung parenchyma and airways on day 2 post infection (Figs. 2D and EV2B); in contrast, both NSP3 mutants had reduced antigen staining in the lungs. Evaluating lung sections from each hamster, NSP3 Mut-1 and Mut-2 had a two- to threefold reduction in antigen score compared to WT at day 2 in the airway, the parenchyma, and overall (Fig. 2E–G, $n = 5$). Similarly, H&E histopathology shows a significant reduction in day 2 cellular infiltration and damage in NSP3 mutant-infected hamsters as compared to WT (Fig. 2H). While WT SARS-CoV-2 infected lungs showed multifocal interstitial pneumonia, perivasculitis, bronchiolitis, and peribronchiolitis, both NSP3 mutants had more focal disease with less extensive damage at day 2. However, at day 4, these trends reversed with both NSP3 mutants showing more antigen staining than WT. Similarly, day 4 and 7 timepoints for H&E showed similar histopathological lesions and damage in both NSP3 mutant- and WT-infected lung sections. Together, the results argue that disrupting NSP3/FMRP binding alters the kinetics of SARS-CoV-2 pathogenesis, despite resulting in similar outcomes.

## NSP3 binds the RNA-binding region of the FMRP KH domains

To understand how the NSP3-FMRP complex contributes to viral infection, we first aimed at obtaining a detailed molecular and structural understanding of the complex. FMRPs are composed of distinct domains, and we therefore constructed a panel of tagged FXR1 fragments and monitored binding to NSP3 1–181 by immunoprecipitation (Fig. 3A). Our results revealed that FXR1 1–215 did not bind NSP3 while FXR1 1–360 did, suggesting that the two central KH domains mediate binding. Indeed FXR1 215–360 comprising the two central KH domains was sufficient for binding to NSP3 1–181. The central KH domains are almost identical in sequence among the FMRPs explaining why we observe

all FMRPs binding to NSP3. Interestingly, the fragile X syndrome disease mutation, I304N, fully blocked the interaction between FXR1 and NSP3 (Fig. EV3A).

To investigate if the interaction is direct, we expressed and purified a number of FXR1 fragments as well as NSP3 1–181 WT and mut-1 mutant. In size-exclusion chromatography experiments, we observed complex formation of FXR1 215–360 and NSP3 1–181 WT (Fig. EV3B). We measured the affinity between a number of recombinant FXR1 fragments and recombinant NSP3 1–181 using isothermal titration calorimetry (ITC) (Fig. EV3C; Appendix Table S2). Consistent with our cellular co-purification experiments, we detected specific binding between NSP3 1–181 and FXR1 215–360 measuring the affinity to 2.3–2.9 µM (Figs. 3B and EV3C; Appendix Table S2). This interaction was abolished by the FXR1 I304N mutation and the NSP3 Mut-1 mutations consistent with the cellular data (Fig. 3B). Importantly, the 20-amino acid region of NSP3 required for interaction was sufficient for binding to FXR1 215–360 with a similar affinity as NSP3 1–181 (Fig. 3B). Collectively this shows that the 20-amino acid sequence of SARS-CoV-2 NSP3 is required and sufficient for interaction with FMRPs. Notably, we confirmed that a reported 23mer NSP3 peptide from alphaviruses binds FXR1 1–122, arguing that these viruses hijack FMPRs by a distinct mechanism (Fig. EV3C) (Kim et al, 2016).

We next sought to define the key residues in NSP3 mediating binding to FMRPs. A five amino acid alanine scan through the SARS-CoV-2 NSP3 peptide did not further narrow down the interaction which argues that multiple residues in this region bind FXR1 (Fig. 3B). We therefore conducted a peptide array experiment where we changed single amino acids to alanine residues in the NSP3 20mer peptide and monitored binding to FXR1 215–360 (Fig. 3C). This study pinpointed three residues in NSP3 critical for binding to FMRPs: Y138, G140, and F145. We subsequently generated a structural model of the complex using AlphaFold multimer (Figs. 3D and EV3D for pLDDT value) (Jumper et al, 2021). Interestingly, this model revealed that the NSP3 peptide interacts with the GxxG motif of the FMRP KH2 domain similarly to how RNA and DNA has been shown to bind KH domains (Ramos et al, 2003) (Fig. 3E). This model further revealed an interaction between FXR1 I304, a residue stabilizing the hydrophobic core of the KH domain, and NSP3 F145 providing an explanation for why mutation of these residues abolish binding. Similarly, Y138 and G140 may play a role in stabilizing a NSP3 loop facilitating further interactions with FMRPs. We have been unable to detect direct binding of RNA to the FXR1 KH domains using a reported RNA that binds full-length FMR1 (Ascano et al, 2012) (Fig. EV4A; Appendix Table S2), suggesting that these KH domains

    

**A**

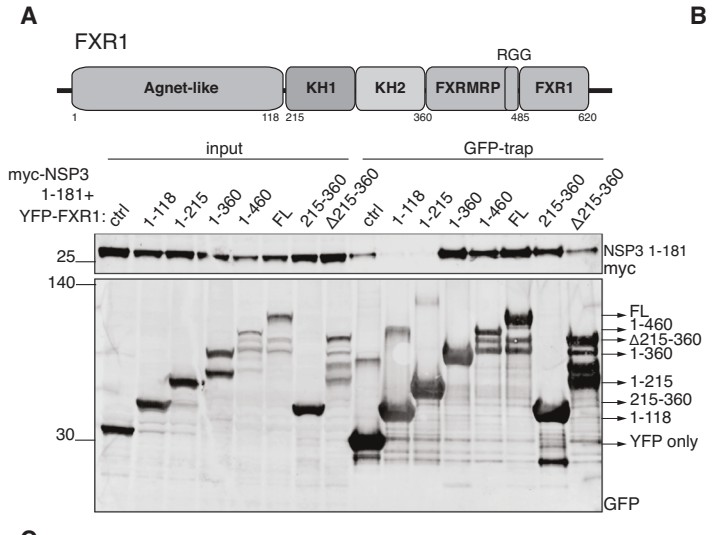

**B**

| Binding affinities of FXR1 215-360 WT | | |
|---|---|---|
| Protein | | $K_D$ (μM) |
| His-NSP3$_{1-181}$ wt $_{129}$...QYEYGTEDDY...$_{138}$ | | 2,9 |
| His-NSP3$_{1-181}$ **Mut-1** $_{129}$...AAAAAAAAAA...$_{138}$ | | NB |
| **Binding affinities of FXR1 215-360 I304N** | | |
| His-NSP3$_{1-181}$ wt $_{129}$...QYEYGTEDDY...$_{138}$ | | NB |
| **Binding affinities of FXR1 215-360 WT** | | |
| Peptide | Sequence | $K_D$ (μM) |
| NSP3$_{wt}$ | QYEYGTEDDYQGKPLEFGATSW | 1,7 |
| NSP3$_{M1}$ | AAAAATEDDYQGKPLEFGATSW | 95,1 |
| NSP3$_{M2}$ | QYEYGAAAAAQGKPLEFGATSW | NB |
| NSP3$_{M3}$ | QYEYGTEDDYAAAAAEFGATSW | NB |
| NSP3$_{M4}$ | QYEYGTEDDYQGKPLAAAAASW | NB |

**C**

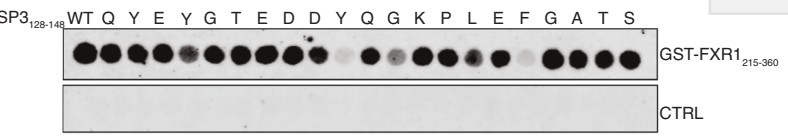

**D**

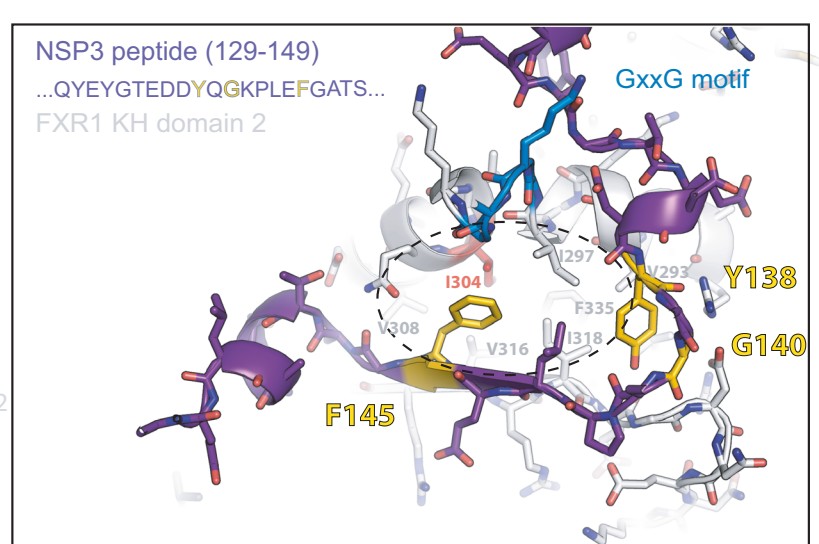

**E**

GxxG motif
conserved Ile
KH domain

KH-DNA KH-RNA FXR1-NSP3 peptide

PDB 2P2R PDB 2ANN AF model

◄ **Figure 3. NSP3 binds to the FMRP KH domains similar to how RNA binds.**

(A) Schematic of FXR1 and immunopurification of the indicated FXR1 fragments and binding to NSP3 1–181 determined. Representative of two biological replicates. (B) Affinity measurements by ITC of the indicated proteins and peptides ($n = 1$). (C) Spot array of the indicated NSP3 peptide incubated with purified FXR1 215–360 to map critical residues required for binding. Representative of two biological replicates. (D) AlphaFold model of the FXR1-NSP3 complex with critical residues in NSP3 indicated by yellow and the I304 residue in FXR1 highlighted. (E) Comparison between KH-DNA and KH-RNA structures and the model of FXR1-NSP3. Source data are available online for this figure.

might be preferentially involved in protein–protein interactions rather than RNA binding. Collectively, our results reveal the key residues in SARS-CoV-2 NSP3 protein that bind to the two central KH domains of FMRPs and suggest RNA mimicry by this peptide.

## NSP3 disrupts the interaction between FMRPs and UBAP2L

To get insight into how the NSP3-FMRP interaction antagonizes antiviral host mechanisms, we determined if NSP3 binding rewires the interactome of FMRPs. Using lysate from cells expressing YFP-FXR1, we added either WT NSP3 peptide or a non-binding control peptide (NSP3$_{M2}$ in Fig. 3B) and then affinity-purified FXR1. We then use mass spectrometry-based proteomics to quantitatively compare the samples. Our results revealed a striking displacement of stress granule components from FXR1 in the presence of the NSP3 WT peptide (Fig. 4A; Dataset EV1). UBAP2L, one of the most affected proteins, shows a strong reduction in co-purification consistent with a possible direct interaction between UBAP2L and FXR1 215–360 shown by a prior two-hybrid screen (Sakai et al, 2011) and suggested by the RNAse resistant interaction observed in immunopurifications (Sanders et al, 2020). To confirm, we conducted the inverse experiment and immunopurified UBAP2L-Venus in the presence of WT NSP3 or mutant NSP3 peptide revealing that the entire FMRP-TDRD3-TOP3B complex was displaced (Fig. 4B; Dataset EV1). Consistent with the FXR1 I304N mutant being unable to bind the NSP3 peptide this mutant was also defective in binding to UBAP2L and stress granule components in cells (Fig. EV4B,C; Dataset EV1). A recently reported FXR1 mutant unable to form cytoplasmic granules, FXR1 L351P, also failed to bind UBAP2L (Kang et al, 2022) (Fig. EV4C). Together, the results argue that NSP3 binding disrupts the interactions between UBAP2L and FMRPs.

The interactome data suggested that UBAP2L and NSP3 compete for binding to a similar interface on FMRPs. To test this directly, we first mapped the binding site in UBAP2L to FMRPs. A truncation analysis of UBAP2L identified the region from 200 to 400 as required for interaction (Fig. EV4D). To further map the site of interaction, we generated a peptide array that covered this region of UBAP2L with 20mer peptides shifted by two amino acids at a time. We observed specific binding of FXR1 215–360 to peptides spanning 243–274 in UBAP2L; these results were further supported by immunopurification of UBAP2L fragments (Fig. 4C,D). An alanine scan through UBAP2L 247–266 pinpointed W249, L253, K257, I258, and F259 as critical residues for binding (Fig. 4D, lower panel). Based on this finding, we generated UBAP2L I258A/F259A and in addition a charge swap mutant, UBAP2L E251K/D252K, which both showed a clear reduction in binding to FXR1 in immunopurifications (Fig. 4E). In contrast mutating the G3BP1/2 binding site in UBAP2L (F518L/F523G) did not affect FXR1

binding (Fig. 4E). This region is conserved in UBAP2 arguing that FMRP interaction mode is conserved between UBAP2L and UBAP2. We measured the affinity of a UBAP2L peptide spanning residues 243–270 which revealed an affinity of 8.5 μM, and we confirmed that the NSP3 peptide and UBAP2L peptide competed for binding to FXR1 215-360 by ITC (Fig. 4F,G; Appendix Table S2).

Consistent with this result, an AlphaFold model of the UBAP2L-FXR1 complex revealed a similar mode of interaction of UBAP2L with the KH domains as that observed with NSP3 (Fig. 4H; Appendix Fig. S1A for pLDDT value). In this model, FXR1 I304 interacts with UBAP2L F259, similar to its interaction with NSP3 F145. We noted several reported phosphorylation sites in the region of UBAP2L binding to FMRPs providing a means to regulate the interaction. To investigate this possibility, we measured the binding affinity of three UBAP2L phosphopeptides to FXR1 by ITC. Interestingly, Thr246 phosphorylation resulted in increased affinity with a Kd of 3.5, while phosphorylation of Ser254 and Ser262 disrupted the interaction (Fig. 4F). The effects of these phosphorylations were consistent with the AlphaFold model of the complex (Fig. 4H; Appendix Fig. S1B). Collectively, these data reveal that SARS-CoV-2 NSP3 competes directly with UBAP2L for binding to FMRPs and displaces the FMRP-TDRD3-TOP3B complex from UBAP2L.

## The incorporation of FMRPs into stress granules is disrupted by NSP3

Our data suggested that the ability of NSP3 to antagonize host cell antiviral mechanisms could be through an effect on stress granule composition and assembly although other proviral functions of the interaction cannot be excluded. To establish if FMRP incorporation into stress granules is affected during infection, we investigated FXR1 localization during infection in VeroE6 cells. We observed that FXR1 associated with stress granules in cells expressing low levels of the SARS-CoV-2 N protein. As the levels of N increased reflecting later stages of infection, FXR1 localization shifted and was evenly distributed throughout the cytoplasm similar to uninfected cells (Fig. EV5A,B). We noted that the total levels of FXR1 increased during infection consistent with a previous study (Stukalov et al, 2021) (Fig. EV5C). Since UBAP2L plays an important role in nucleating stress granules, we speculated that NSP3 affected the ability of FXR1 to associate with these structures through competition during infection. Consistent with this idea FXR1 I304N and FXR1 L351P were unable to form stress granules induced by arsenite (Fig. EV5D). To test this directly, we expressed NSP3 1–181 WT or NSP3 Mut-2 and monitored the ability of endogenous FXR1 to associate with stress granules following their induction by arsenite (Fig. 5A,B). In line with our interaction data, we observed that FXR1 was impaired in associating with stress

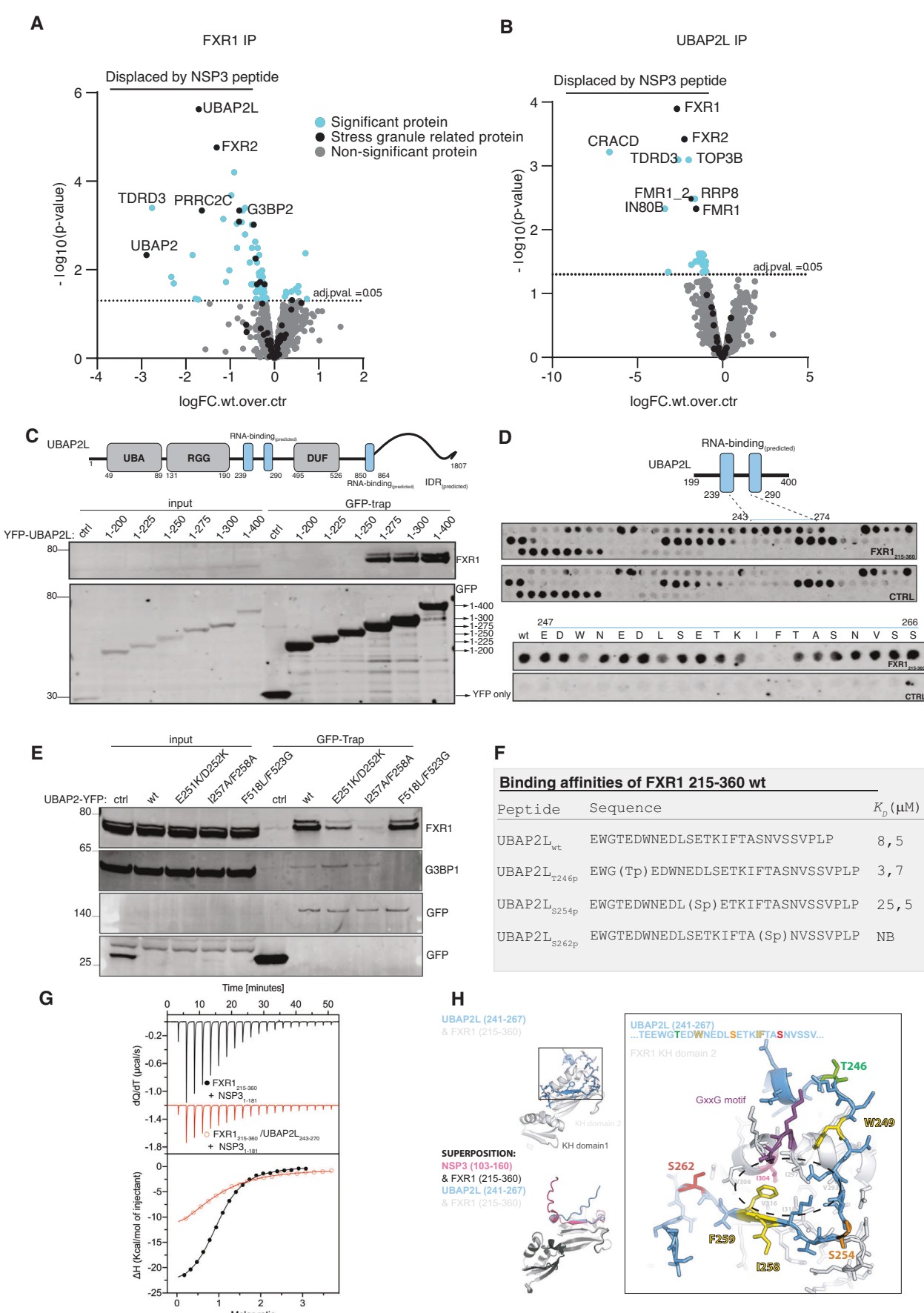

**Figure 4.  NSP3 disrupts the UBAP2L-FMRP complex.**

(A) FXR1 was affinity-purified and incubated with either WT NSP3 or mutant NSP3 peptide and interactomes determined by MS to determine proteins specifically displaced by WT NSP3. Data from four technical replicates. (B) As (A) but using UBAP2L as a bait. Data from four technical replicates. (C) Schematic of UBAP2L and truncation analysis to identify FXR1 binding site ($n = 1$). (D) Peptide array of UBAP2L 199–400 to identify FXR1 binding region and lower part single Ala scan through UBAP2L 247–266 to identify critical residues ($n = 1$). (E) Immunopurification of indicated UBAP2L constructs to determine binding to FXR1 and G3BP1. Representative of two biological replicates. (F) ITC measurements of indicated UBAP2L peptides to FXR1 215–360 ($n = 1$). (G) Competition between NSP3 peptide and UBAP2L peptide for binding to FXR1 215–360. The black trace is NSP3 binding to FXR1 while the red trace is NSP3 binding to FXR1 preincubated with UBAP2L peptide ($n = 1$). (H) Alphafold model of the FXR1-UBAP2L complex highlighting critical residues in yellow and phosphorylation sites. Data information: In (A, B), the statistical comparison was done using a two-sided unpaired $t$ test. Source data are available online for this figure.

granules in the presence of NSP3 WT, but not NSP3 Mut-2. Importantly this effect appeared to be specific to FXR1 as G3BP1 incorporation was not strongly affected by NSP3. Using live-cell imaging to monitor the incorporation of YFP-FXR1 and YFP-G3BP1 into stress granules, we observed a strong reduction of FXR1 incorporation when co-expressed with Cherry-tagged NSP3 1–181 WT (Appendix Fig. S2A) contrasting a small reduction with G3BP1. Similarly, we complemented HeLa UBAP2L KO cells (Youn et al, 2018) with our UBAP2L mutants and analyzed incorporation of FXR1 into stress granules upon arsenite addition. Preventing the interaction between UBAP2L and FMRPs did not affect the ability of UBAP2L to form stress granules in contrast to the UBAP2L mutant unable to bind G3BP1/2 (Appendix Fig. S2B). Incorporation of FXR1 into stress granules was strongly impaired in the UBAP2L KO cell lines as expected but could be restored by expressing UBAP2L-YFP. However, mutations in UBAP2L preventing FXR1 binding also prevented efficient incorporation of FXR1 into stress granules (Fig. 5C,D). Collectively, our results show that NSP3 blocks the UBAP2L-FMRP interaction necessary for association of FMRPs with stress granules and this could act to antagonize antiviral defense mechanisms efficiently during early stages of infection.

## Discussion

Here, we uncover a novel mechanism by which SARS-CoV-2 NSP3 disrupts the UBAP2L-FMRP interaction which could be linked to antagonizing stress granule assembly. Specifically, a 20-amino acid region of the NSP3 hypervariable region, well conserved in Sarbeco coronaviruses, mediates binding to FMRPs hereby competing with binding to UBAP2L and preventing FMRP incorporation into stress granules. This mechanism is reminiscent of how SARS-CoV-2 N protein antagonizes binding of G3BP1/2 to stress granule components through a ΦxFG motif. Using two NSP3 mutants that disrupt parts of this 20-amino acid region, we found attenuated virus replication in vitro not driven by type I interferon activity. However, both NSP3 mutants had a modest impact on in vivo SARS-CoV-2 infection, causing only a delay in the kinetics of pathogenesis. One possibility for the modest effect is that the role of N and NSP3 on stress granule biology is to some extend redundant. Overall, our study demonstrates that the NSP3 interaction with FMRP through its hypervariable region impacts SARS-CoV-2 infection both in vitro and in vivo. This NSP3 interaction with FMPR may be critical in delaying stress granule formation, thus blunting antiviral effects and facilitating improved coronavirus infection.

Stress granule formation is a highly complex process with numerous components, cell-type specificity, and critical functions

in host responses including antiviral defense. Importantly, viruses have developed numerous approaches to disrupt and hijack stress granule elements to facilitate viral infection. For SARS-CoV-2, the N protein powerfully disrupts stress granule formation using its ΦxFG motif to prevent G3BP1/2 interactions (Kruse et al, 2021). However, the multifunctional N protein has numerous activities during infection including viral RNA replication (Almazan et al, 2004), RNA transcription (Almazan et al, 2004; Wu et al, 2014; Zuniga et al, 2010), and antagonizing innate immunity (Zhao et al, 2021). Therefore, N-mediated stress granule antagonism is likely delayed until sufficient N protein accumulates late during infection. While the N protein eventually controls stress granules via its G3BP1/2 interaction, NSP3 binding to FMPRs could interfere with stress granule formation and antiviral activities early during infection.

To our knowledge, SARS-CoV-2 NSP3 is the first example of a virus targeting UBAP2L-FMRP interactions. Interestingly, the required peptide sequence in NSP3 is more extensive than just the ΦxFG found in SARS-CoV-2 N. In our AlphaFold models, both the NSP3 peptide and the UBAP2L peptide engage the FMRP KH domain at a similar site as RNA in reported KH-RNA structures. This is the first example of a KH domain binding to a peptide, suggesting that this protein fold is more versatile in function (Ramos et al, 2003). The KH domain of FMRPs are known to bind specific RNAs and disruption of its stress granule incorporation may prevent sequestration of RNAs key to supporting viral replication. At present, we cannot exclude that the interaction between NSP3 and FMRPs is unrelated to an effect on stress granules and that it has a proviral function that needs to be elucidated. As an example, FMRPs may be highjacked by SARS-CoV-2 NSP3 to play a role in viral transcription or translation. To discriminate between a role in stress granule biology or a proviral function, it will be critical to conduct infection assays in FMRP or UBAP2L knockout cell lines in future experiments.

Together with the N protein, NSP3 antagonism demonstrates the significant genetic capital SARS-CoV-2 uses to modulate stress granule functions. Given that NSP3 binds to N through its Ubl domain, both stress granule disruptive viral proteins would be in close proximity. This viral complex could ensure efficient stress granule disassembly at the sites where nascent viral RNA emerges from the double- membrane vesicles through pores formed by NSP3 (Wolff et al, 2020). Importantly, disruption of either NSP3 or N function on stress granules has implications for SARS-CoV-2 infection and pathogenesis. Therefore, the importance of controlling stress granules may open therapeutic approaches to target these interfaces to facilitate direct antiviral drug treatments and live-attenuated vaccine approaches. However, similar interactions with host proteins increase the possibility of toxicity and off-target effect. Overall, the interactions of both NSP3 and N proteins with

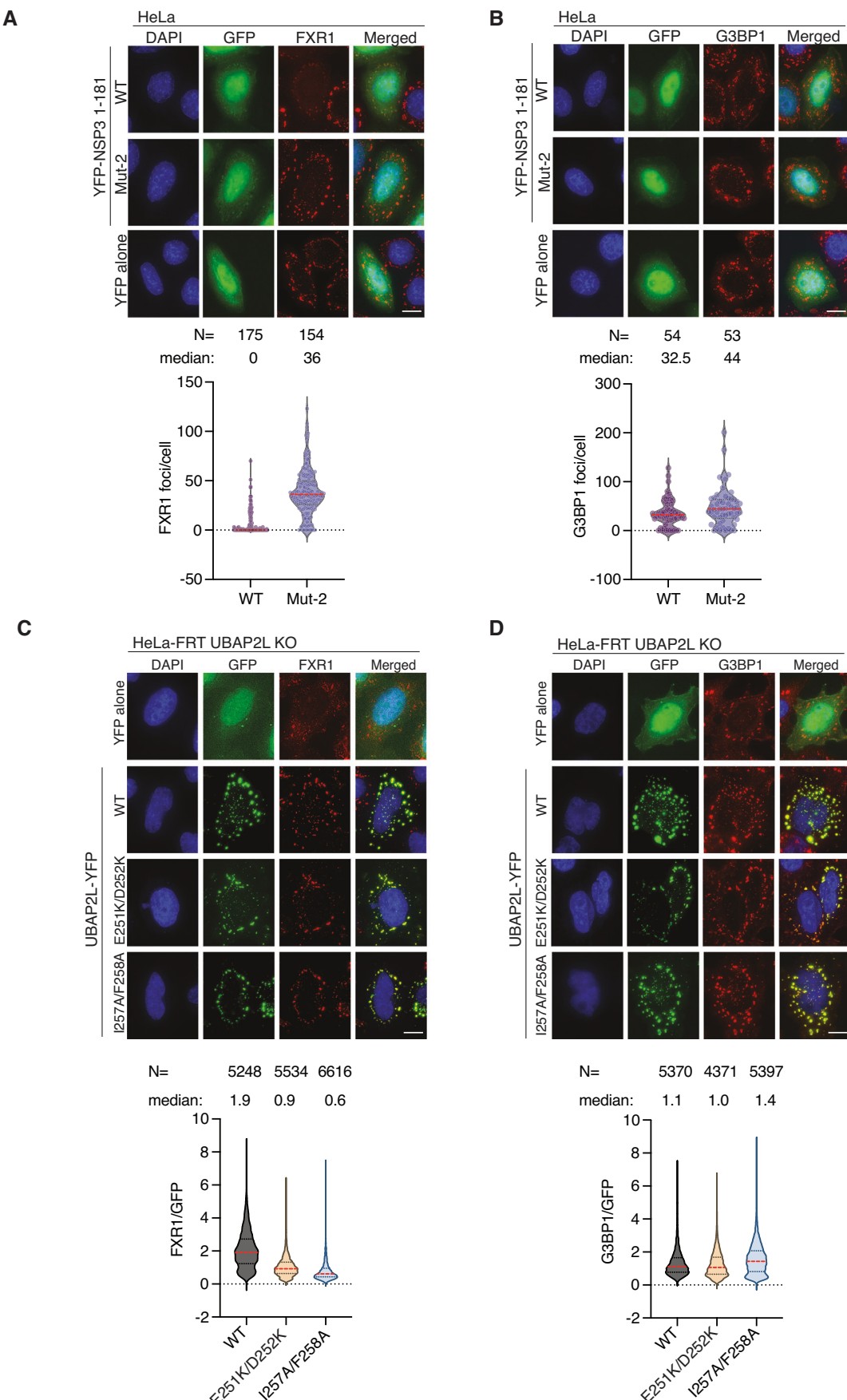

Figure 5.   NSP3 prevents the incorporation of FMRPs into stress granules.

(A) Analysis of FXR1 stress granule association in the presence of NSP3 WT or NSP3 A4. HeLa cells were treated 30 min with arsenite before fixation and number of FXR1 foci quantified per cell. Only cells expressing NSP3 was analyzed. (B) Similar as (A) but number of G3BP1 foci analyzed. (C) HeLa UBAP2L KO cells were complemented with UBAP2L-YFP constructs and cells treated with arsenite for 30 min before fixation. The fluorescence intensity of FXR1 to UBAP2L (GFP signal) was quantified. (D) As in (C) but staining for G3BP1. (A–D) representative stills from the immunofluorescence is shown with a scale bar of 10 µM indicated in lower left corner. Combined data from three and two biological replicates is shown in (A, B), respectively. (C, D) A pool of four biological replicates is shown in the graphs. Data information: Violin plots (A–D) with all points shown (A, B) with the median indicated with red line. The number of cells (A, B) or foci (C, D) analyzed per condition is indicated above the plot. Scale bars represent 10 µm. Source data are available online for this figure.

stress granule components is a critical interface that dictates infection outcome after SARS-CoV-2 infection.

In addition to providing new insight into SARS-CoV-2 biology, our work also hints at potential underlying molecular defects of fragile X syndrome. Recent data suggest that brain development is linked to stress granule function, but understanding this will require deeper mechanistic insight into stress granule biology. The molecular detailed understanding of the FMRP-UBAP2L complex described here can potentially help explain its role in brain development and how this is affected by disease mutations such as FMR1 I304N. It will furthermore be interesting to explore if antiviral defense mechanisms are affected in patients with fragile X syndrome or harboring mutations in stress granule components and whether this affects the disease trajectory of COVID19.

## Methods

### Cloning

NSP3, FXR1, and UBAP2L fragments were cloned by PCR amplification and restriction digest cloning into pcDNA5/FRT/TO/FRT YFP using BamHI/BglII and NotI. Ten Ala mutations in SARS-CoV-2 NSP3 1–181 and N-terminal NSP3 fragments from coronaviruses were obtained as gene synthesized fragments (Thermo Scientific) and restriction enzyme cloned using BamHI and NotI into pcDNA5/FRT/TO YFP. UBAP2L-YFP is described in Kruse et al, 2021 and mutations in this construct and FXR1 are achieved through quick change PCR. For primer sequences, see Appendix Table S1.

### Cell culture

HeLa, HEK293 and HeLa-FRT parental, and HeLa-FRT UBAP2L KO cell lines were cultured in DMEM GlutaMax media (Thermo Fisher Scientific) at 37 °C in a humidified incubator with 5% CO$_2$. The media was supplemented with 10% FBS (HyClone) and 1% PenSrep (Thermo Fisher Scientific). The cell lines were not authenticated or checked for mycoplasm during the work. HeLa and HEK293 cells were from ATCC (CCL-2 and CRL-1573) while HeLa-FRT/TRex UBAP2L KO cells were a kind gift from Anne-Claude Gingras and described in (Youn et al, 2018).

VeroE6 cells were grown in high glucose DMEM (Gibco #11965092) with 10% fetal bovine serum and 1× antibiotic-antimycotic. TMPRSS2-expressing VeroE6 cells were grown in low glucose DMEM (Gibco #11885084) with sodium pyruvate, 10% FBS, and 1 mg/mL Geneticin™ (Invitrogen #10131027). VeroE6 cells were derived from a lab stock and have not been authenticated. Calu3 2B4 cells were grown in high glucose DMEM

(Gibco #11965092) with 10% defined fetal bovine serum, 1 mM sodium pyruvate, and 1× antibiotic-antimycotic. Calu3 2B4 cells were originally provided as a gift from Dr. Kent Tseng, have been lab passaged for several years, and not been authenticated. Both VeroE6 cells and Calu3 2B4 cells are tested periodically for mycoplasma were shown to be negative upon the last test (07/2023).

### Viruses

The SARS-CoV-2 infectious clones were based on the USA-WA1/2020 sequence provided by the World Reference Center of Emerging Viruses and Arboviruses and the USA Centers for Disease Control and Prevention (Harcourt et al, 2020). Mutant viruses (NSP3 Mut-11 and NSP3 Mut-2) were generated with restriction enzyme-based cloning using gBlocks encoding the mutations (Integrated DNA Technologies) and our reverse genetics system as previously described (Xie et al, 2021a; Xie et al, 2020). Virus stock was generated in TMPRSS2-expressing VeroE6 cells to prevent mutations from occurring at the FCS, as previously described (Johnson et al, 2021). Viral RNA was extracted from virus stock and cDNA was generated to verify mutations by Sanger sequencing.

### Biosafety

The synthetic construction of SARS-CoV-2 NSP3 mutants was reviewed for DURC/P3CO policies and approved by the University of Texas Medical Branch Institutional Biosafety Committee. All studies in animals were conducted under a protocol approved by the UTMB Institutional Animal Care and Use Committee and complied with USDA guidelines in a laboratory accredited by the Association for Assessment and Accreditation of Laboratory Animal Care. UTMB is a registered Research Facility under the Animal Welfare Act. It has a current assurance (A3314-01) with the Office of Laboratory Animal Welfare (OLAW), in compliance with NIH Policy. Procedures involving infectious SARS-CoV-2 were performed in the Galveston National Laboratory ABSL3 facility.

### In vitro infection

Vira infections in VeroE6 and Calu3 2B4 were carried out as previously described (Vu et al, 2022). Briefly, growth media was removed, and cells were infected with WT or mutant SARS-CoV-2 at an MOI of 0.01 for 45 min at 37 °C with 5% CO$_2$. After absorption, cells were washed three times with PBS, and fresh complete growth media was added. Three or more biological replicates were collected at each time point and each experiment was performed at least twice. Samples were titrated with plaque assay or focus-forming assays.

## Virus quantitation via focus-forming assay

Focus-forming assays (FFAs) were performed as previously described (Johnson et al, 2022). Briefly, VeroE6 cells were seeded in 96-well plates to be 100% confluent. Samples were serially diluted in serum-free media and 20 μl was used to infect cells. Cells were incubated for 45 min at 37 °C with 5% $CO_2$ before 0.85% methylcellulose overlay was added. Cells were incubated for 24 h at 37 °C with 5% $CO_2$. After incubation, the overlay was removed, and cells were washed three times with PBS before fixed and virus inactivated by 10% formalin for 30 min at room temperature. Cells were then permeabilized and blocked with 0.1% saponin/0.1% BSA in PBS before incubated with α-SARS-CoV-2 Nucleocapsid primary antibody (Cell Signaling Technology, #68344) at 1:1000 in permeabilization/blocking buffer overnight at 4 °C. Cells were then washed (3x) with PBS before incubated with Alexa Fluor™ 555-conjugated α-mouse secondary antibody (Invitrogen #A28180) at 1:2000 in permeabilization/blocking buffer for 1 h at room temperature. Cells were washed (3×) with PBS. Fluorescent foci images were captured using a Cytation 7 cell imaging multi-mode reader (BioTek), and foci were counted manually.

## Hamster infection

Three- to four-week-old male golden Syrian hamsters (HsdHan:AURA strain) were purchased from Envigo. Upon receipt, all animals were housed in sterile ventilated cages within a specific pathogen-free environment until infection. Hamsters were intranasally infected with $10^5$ pfu of WT or NSP3 Mut-1 or NSP3 Mut-2 SARS-CoV-2 in 100 μl. Infected hamsters were weighed and monitored for illness over 7 days. Hamsters were anesthetized with isoflurane and nasal washes were collected with 400 μl of PBS on endpoint days (2, 4, and 7 dpi). Hamsters were euthanized by $CO_2$ for organ collection. Nasal wash and lung were collected to measure viral titer and RNA. Left lungs were collected for histopathology.

## Histology

Left lung lobes were harvested from hamsters and fixed in 10% buffered formalin solution for at least 7 days. Fixed tissue was then embedded in paraffin, cut into 5-μm sections, and stained with hematoxylin and eosin (H&E) on a SAKURA VIP6 processor by the University of Texas Medical Branch Surgical Pathology Laboratory. For antigen staining, paraffin-embedded sections were warmed at 56 °C for 10 min, deparaffinized with xylene (3 × 5-min washes) and graded ethanol (3 × 100% 5-min washes, 1 × 95% 5-min wash), and rehydrated in distilled water. After rehydration, antigen retrieval was performed by steaming slides in antigen retrieval solution (10 mM sodium citrate, 0.05% Tween-20, pH 6) for 40 min (boil antigen retrieval solution in microwave, add slides to boiling solution, and incubate in steamer). After cooling and rinsing in distilled water, endogenous peroxidases were quenched by incubating slides in TBS with 0.3% $H_2O_2$ for 15 min followed by 2 × 5-min washes in 0.05% TBST. Sections were blocked with 10% normal goat serum in BSA diluent (1% BSA in 0.05% TBST) for 30 min at room temperature. Sections were incubated with primary anti-N antibody (Sino #40143-R001) at 1:1000 in BSA diluent overnight at 4 °C. Following overnight primary antibody

incubation, sections were washed 3× for 5 min in TBST. Sections were incubated in secondary HRP-conjugated anti-rabbit antibody (Cell Signaling Technology #7074) at 1:200 in BSA diluent for 1 h at room temperature. Following secondary antibody incubation, sections were washed 3× for 5 min in TBST. To visualize antigen, sections were incubated in ImmPACT NovaRED (Vector Laboratories #SK-4805) for 3 min at room temperature before rinsed with TBST to stop the reaction followed by 1 × 5-min wash in distilled water. Sections were incubated in hematoxylin for 5 min at room temperature to counterstain before rinsing in water to stop the reaction. Sections were dehydrated by incubating in the previous xylene and graded ethanol baths in reverse order before mounted with coverslips. Viral antigen staining was scored blinded on a scale of 0 (none) to 3 (most) in 0.25 score increments with scores average from at least two lung sections from each hamster.

## Immunofluorescence

Hela or HeLa-FRT UBAP2L KO or parental cells were seeded in six-well dishes with coverslips at 25% confluency. Cells were transfected the day after with 250 ng DNA and 1 μL or jet OPTIMUS (Polyplus) reagent overnight in DMEM media supplemented with FBS (10%) (HyClone) and PenStrep (1%) (Thermo Scientific). Media was changed to media containing 0.5 μM sodium arsenite for 30 min to induce stress granules formation. After washing with PBS cells were fixed for 20 min with 4% formaldehyde in PBS. Cells were premeballized for 10 min with PBS 0.5% Triton-100. Following three 5-min washes with PBS-T (0.05% Tween), 25 mM Glycine was incubated overnight at 4 °C. Coverslips were blocked in TBST (0.05% Tween) 3% BSA for 45 min at room temperature. Primary antibodies (anti-G3BP1 mouse abcam #ab56574, anti-FXR1 mouse clone 6BG10 Milipore #05-1529, GFP booster atto488 (Chromotek)) were incubated at 1:400 dilution in 3% BSA TBST (0.05% Tween) overnight at 4 °C. Following three 5-min washes with TBST (0,05% Tween), coverslips were incubated with secondary antibodies for 1 h at room temperature. Coverslips were mounted in MOWIOL mounting solution (Calbiochem #475904) and imaged on a Delta-Vision Elite microscope (DeltaVision) with 60x oil objective. Data were analyzed in Fiji and plotted with a Prism 9 GraphPad software.

## Live-cell imaging

HeLa cells were seeded at 25% confluency six-well dishes and transfected on the following day with 200 ng DNA and 1 μL JetOptimus reagent overnight in 3 mL media. Media was changed on the next day, and cells were seeded in eight-well ibidi dishes at 40% confluency. Pictures were taken every 5–10 min in 2 z-stacks on a Delta-Vision Elite microscope (DeltaVision) with 60× oil objective. Cells were filmed 48 h post transfection for 10 min and sodium arsenite was added to a final concentration of 0.5 μM. Cells were filmed for additional hours to follow the formation of stress granules. Data were analyzed in Fiji and plotted with a Prism 9 GraphPad software.

## Immunoprecipitations

Cells were seeded at 25% confluency in 15-$cm^3$ dishes with 2 μg DNA and 2 μL JetOptimus reagent overnight in 15 mL media.

Media was changed on the following day, and cells were collected after 48 or 24 h post transfection. Cell pellets from each 15-cm³ dish were lyzed in 350 μL lysis buffer (100 mM NaCl, 50 mM Tris pH 7.4, 0.1% NP40, 0.2% Triton-100, 1 mM DTT) supplemented with protease (complete mini EDTA-free, Roche) and phosphatase inhibitor tablets (Roche). Lysate was sonicated at 4 °C for ten cycles of 30 s ON, 30 s OFF using a Bioruptor sonicator. Following sonication lysates were cleared for 45 min at 20,000× g. Supernatants were collected and concentrations were measured. Lysates were incubated with 10 μL pre-equilibrated GFP-trap beads for 1 h at 4 °C on a rotor wheel. Beads were washed three times with 800 μL wash buffer (150 mM NaCl, 50 mM Tris pH 7.4, 0.05% NP40, 5% glycerol, 1 mM DTT). If immunoprecipitations were prepared for mass spectrometry analysis, one additional wash with basic wash buffer (100 mM NaCl, 50 mM Tris pH 7.4, 5% glycerol) was performed. If immunoprecipitations were analyzed by SDS-PAGE and western blot, 25 μL 2× LSB (Thermo Fisher Scientific) was used to elute the samples. Samples were then boiled and separated by SDS-PAGE and processed for western blot using the following antibodies: FMR1 rabbit (Sigma Aldrich – HPA050118), FXR1 mouse (Santa Cruz 374148), c-myc (Santa Cruz 9E10), GFP rabbit (made in-house against GFP), G3BP1 (Cell Signaling Technology 17798S), UBAP2L rabbit (Bethyl A300-533A).

## Size-exclusion chromatography

Recombinant proteins (GST-NSP3 1–181 and His-FXR1 215–360) were run on a Supperose 200 column (Cytiva) and an AKTA system. For analyzing direct protein–protein interactions, proteins were pre-mixed for 30 min on ice, and then following a 30-s spin at 20,000× g on a table-top centrifuge, they were run on the column. 500 μL fractions were collected, and peak fractions were analyzed by SDS-PAGE.

## Structural modeling

Structures of complexes between human FXR1 and NSP3 or UBA2PL were predicted with Alphafold multimer (Evans, 2023; Jumper et al, 2021; Wilson et al, 2022) based on full-length amino acid sequences for human FXR1 (UniProt entry A0A0F7L1S3) and human UBA2PL (Uniprot entry F8W726) and SARS-CoV-2 NSP3 residues 103–161 (Uniprot entry P0DTD1).

Phosphorylations of serine and threonine residues were modeled and locally geometry-refined in Coot (Emsley and Cowtan, 2004). All structural models/PDBs and their pLDDT scores were visualized in PyMOL (Schrodinger, LLC).

## Isothermal titration calorimetry (ITC)

Peptides were purchased from Peptide 2.0 Inc. (Chantilly. VA, USA). The purity obtained in the synthesis was 95–98% as determined by high-performance liquid chromatography (HPLC) and subsequent analysis by mass spectrometry. Prior to ITC experiments both the proteins and the peptides were extensively dialyzed against 50 mM sodium phosphate pH 7.5, 150 mM NaCl, 0.5 mM TCEP. All ITC experiments were performed on an Auto-iTC200 instrument (Microcal, Malvern Instruments Ltd.) at 25 °C. Both peptide and protein concentrations were determined using a spectrometer by measuring the absorbance at 280 nm and applying values for the extinction

coefficients computed from the corresponding amino acid sequences by the ProtParam program (http://web.expasy.org/protparam/). FXR1 constructs (FXR1[212–289], FXR1[215–360]) and NCAP_SARS2[1–419] at ~300 μM or 100 μM (FXR1[1–122]) concentration were loaded into the syringe and titrated into the calorimetric cell containing NSP3 1–181 at ~20 μM or ~10 μM, respectively. NSP3 128–148 peptide (and variants) and UBAP2L 243–270 (and variants) at approximately 300 μM were loaded into the syringe and titrated into the calorimetric cell containing FXR1[215–360] at ~20 μM. For competition experiments, NSP3 1–181 at ~300 μM concentration was loaded into the syringe and titrated into the calorimetric cell containing either FXR1[215–360] or FXR1[215–360] saturated with UBAP2L 243–270 at ~20 μM FXR1[215–360] concentration. The reference cell was filled with distilled water. In all assays, the titration sequence consisted of a single 0.4 μl injection followed by 19 injections, 2 μl each, with 150 s spacing between injections to ensure that the thermal power returns to the baseline before the next injection. The stirring speed was 750 rpm. Control experiments with the FXR1, NCAP_SARS2 constructs or the NSP3 and UBAP2L peptides injected in the sample cell filled with buffer were carried out under the same experimental conditions. These control experiments showed heats of dilution negligible in all cases. The heats per injection normalized per mole of injectant versus the molar ratio [titrant in syringe]/[titrand in calorimetric cell] were fitted to a single-site model. RNA sequence for RNA used in S3A: rGrGrArUrCrArUrUrUrUrGrUrUrGrGrArCrUrCrArArUrUrUrCrArrArCrUrCrUrArArArCrUrUrUrArArArCrUrUrUrGrCrArUrUrGrGrUrUrGrGrArCrArCrCrU. Data were analyzed with MicroCal PEAQ-ITC (version 1.1.0.1262) analysis software (Malvern Instruments Ltd.).

## Protein production and purification

The NSP3 and FXR1 fragments were expressed in the *E. coli* strain BL21(DE3) overnight at 18 °C. Cells were harvested and resuspended in 50 mM NaP pH = 7.5; 300 mM NaCl; 10% glycerol; 0.5 mM TCEP; 1× Complete EDTA-free tablets (Roche) (and 10 mM imidazole for His-tag purifications) and lysed with high-pressure-homogonizer (Avestin) and cell extract clarified by centrifugation. The clarified lysate was loaded onto a His-tag or GST-tag affinity column, and following washing with resuspension buffer the proteins were eluted with either an imidazole gradient or glutathione-containing buffer. Peak fractions were pooled and further purified on a size-exclusion chromatography column preequilibrated with 50 mM NaP pH = 7.5; 150 mM NaCl; 10% glycerol; 0.5 mM TCEP.

## FXR1 foci during SARS-CoV-2 infection

VeroE6 cells were infected with SARS-CoV-2 (SARS-CoV-2/01/human/2020/SWE accession no/GeneBank no MT093571.1) and fixed in 4% formaldehyde at 3 and 6 h post infection. Then cells were quenched with 10 mM glycine, and permeabilized with PBS and 0.5% Triton X-100, and incubated with primary antibodies against SARS-CoV-2 nucleocapsid ((1:500) Sino Biological Inc., 40143-R001) and FXR1 ((1:500) MerckMillipore, 05-1529) followed by incubation with conjugated secondary antibodies anti-rabbit Alexa555 and antimouse Alexa488 (1:500, Thermo Fisher Scientific). Thereafter cells were stained with an APC-conjugated antibody directed against dsRNA J2 ((1:200) Scicons 10010500, the antibody was conjugated using APC Conjugation Kit - Lightning-Link® (ab201807)). Nuclei

were counterstained with DAPI (diluted 1:1500). For immunofluorescence, images were obtained using Leica SP8 Laser Scanning Confocal Microscope with a ×63 oil objective (Leica) and Leica Application Suit X software (LAS X, Leica). In order to quantify FXR1, foci images were acquired using an Olympus CKX53 microscope using a ×20 objective (Olympus). An area outline was drawn for each cell and the total fluorescent signal of nucleoprotein and amount of FXR1/area was counted using "analyze particles" in ImageJ/Fiji. The threshold was set equal for all measurements and cells with saturated signal was excluded. All data were adjusted for background signal.

## Affinity purification and mass spectrometry (AP-MS)

Partial on-bead digestion was used for peptide elution from GFP-Trap Agarose (Chromotek). Briefly, 100 µl of elution buffer (2 M urea; 2 mM DTT; 20 µg/ml trypsin; and 50 mM Tris, pH 7.5) was added and incubated at 37 °C for 30 min. Samples were alkylated with 25 mM CAA and digested overnight at room temperature before addition of 1% trifluoroacetic acid (TFA) to stop digestion. Peptides were desalted and purified with styrene–divinylbenzene reversed-phase sulfonate (SDB-RPS) StageTips. Briefly, two layers of SDB-RPS were prepared with 100 µl wash buffer (0.2% TFA in $H_2O$). Peptides were loaded on top and centrifuged for 5 min at $500× g$, and washed with 150 µl wash buffer. Finally, peptides were eluted with 50 µl elution buffer (80% ACN and 1% ammonia) and vacuum-dried. Dried peptides were dissolved in 2% acetonitrile (ACN) and 0.1% TFA in water and stored at −20 °C.

## LC-MS analysis

Liquid chromatography-mass spectrometry (LC-MS) analysis was performed with an EASY-nLC-1200 system (Thermo Fisher Scientific) connected to a trapped ion mobility spectrometry quadrupole time-of-flight mass spectrometer (timsTOF Pro, Bruker Daltonik GmbH, Germany) with a nano-electrospray ion source (Captive spray, Bruker Daltonik GmbH). Peptides were loaded on a 50 cm in-house packed HPLC-column (75-µm inner diameter packed with 1.9-µm ReproSilPur C18-AQ silica beads, Dr. Maisch GmbH, Germany). Peptides were separated using a linear gradient from 5 to 30% buffer B (0.1% formic acid, 80% ACN in LC-MS grade H2O) in 43 min followed by an increase to 60% buffer B for 7 min, then to 95% buffer B for 5 min and back to 5% buffer B in the final 5 min at 300 nl/min. Buffer A consisted of 0.1% formic acid in LC-MS grade H2O. The total gradient length was 60 min. We used an in-house made column oven to keep the column temperature constant at 60 °C.

Mass spectrometric analysis was performed essentially as described in Brunner et al (Brunner et al, 2022) in data-dependent (ddaPASEF) mode. For ddaPASEF, 1 MS1 survey TIMS-MS and 10 PASEF MS/MS scans were acquired per acquisition cycle. Ion accumulation and ramp time in the dual TIMS analyzer was set to 100 ms each and we analyzed the ion mobility range from $1/K0 = 1.6$ Vs cm$^{-2}$ to 0.6 Vs cm$^{-2}$. Precursor ions for MS/MS analysis were isolated with a 2 Th window for $m/z < 700$ and 3 Th for $m/z > 700$ in a total $m/z$ range of 100–1.700 by synchronizing quadrupole switching events with the precursor elution profile from the TIMS device. The collision energy was lowered linearly as a function of increasing mobility starting from 59 eV at $1/K0 = 1.6$ VS cm$^{-2}$ to 20 eV at $1/K0 = 0.6$ Vs cm$^{-2}$. Singly charged precursor ions were excluded with a polygon filter (otof control, Bruker Daltonik GmbH). Precursors for MS/MS were picked at an intensity threshold of 1.000 arbitrary units (a.u.) and resequenced until reaching a "target value" of 20.000 a.u taking into account a dynamic exclusion of 40 s elution.

For a subset of the samples, we analyzed them on an Orbitrap Exploris™ 480 Mass Spectrometer (Thermo Fisher Scientific), in DDA mode. For DDA analysis, the mass spectrometer was operated in "top-10" data-dependent mode, in which MS spectra were collected in the Orbitrap mass analyzer (300–1650 $m/z$ range, 60,000 resolution) with an automatic gain control (AGC) target of 3E6 and a maximum ion injection time of 60 ms. The most intense ions from the full scan were isolated with an isolation with of 1 $m/z$. After higher-energy collisional dissociation (HCD) at a normalized collision energy (NCE) of 28, MS/MS spectra were collected in the Orbitrap (15,000 resolution) with an AGC target of 1E5 and a maximum ion injection time of 50 ms. Precursor dynamics exclusion was enabled with a duration of 30 s.

## Data analysis of proteomic raw files

Mass spectrometric raw files acquired in ddaPASEF mode were analyzed with MaxQuant (version 1.6.7.0) (Cox and Mann, 2008; Prianichnikov et al, 2020). The Uniprot database (2019 release, UP000005640_9606) was searched with a peptide spectral match (PSM) and protein level FDR of 1%. A minimum of seven amino acids was required including N-terminal acetylation and methionine oxidation as variable modifications and cysteine carbamidomethylation as fixed modification. Enzyme specificity was set to trypsin with a maximum of two allowed missed cleavages. The first and main search mass tolerance was set to 70 ppm and 20 ppm, respectively. Peptide identifications by MS/MS were transferred by matching four-dimensional isotope patterns between the runs (MBR) with a 0.7-min retention-time match window and a 0.05 1/K0 ion mobility window. Label-free quantification was performed with the MaxLFQ algorithm (Cox et al, 2014) and a minimum ratio count of two.

Orbitrap Exploris generated raw data was analyzed using the AlphaPept Search engine (Strauss, 2021) version 0.4.8 using default settings.

## Bioinformatic analysis

Proteomics data analysis was performed with Perseus (Tyanova et al, 2016) and within the R environment (https://www.r-project.org/). MaxQuant output tables were filtered for "Reverse", "Only identified by site modification", and "Potential contaminants" before data analysis. Missing values were imputed after stringent data filtering and based on a normal distribution (width = 0.3; downshift = 1.8) prior to statistical testing. For pairwise proteomic comparisons (two-sided unpaired $t$ test), we applied a permutation-based FDR of 5% to correct for multiple hypothesis testing, including an $s_0$ value (Tusher et al, 2001) of 0.1.

## Experimental study design and statistics

No predetermined estimates of sample size, randomization or blinding were used for biochemical assays and in vitro

immunoprecipitation assays. Different orthogonal assays were used to confirm the main results in addition to the replication of data. For immunofluorescence analysis of stress granule foci, a semi-automatic quantification in Fuji was used to minimize bias. The immunofluorescence results were supported by an orthogonal live-cell imaging approach. We only excluded experiments where the controls did not work.

For virus experiments, replication kinetics in Vero and Calu3 cells were performed in two independent experiments, each in triplicate. Animal experiments were done once according to the results of the power analysis, which is the maximum number of animals approved by the UTMB IACUC in accordance with the principle of reduction. Viral loads were measured via independent methods (focus-forming assays) for cell culture, primary tissue, and animal experiments. Hamsters were received from Envigo by dedicated animal research personnel at UTMB, who randomly assigned the hamsters to cages of three or five animals with no additional knowledge of study design. No further randomization was performed by research personnel. Blinding was not possible due to safety considerations regarding infected animals and cell culture. A two-tailed students $T$ test was used to determine changes between measured parameters for the different viruses which is an appropriate test.

## Reagent availability

All DNA constructs generated in this project are available upon request to JN. Recombinant wild-type and mutant SARS-CoV-2 described in this manuscript will be made available through the World Reference Center for Emerging Viruses and Arboviruses (WRCEVA) at UTMB through material transfer agreement.

# Data availability

The raw mass spectrometry data are available via ProteomeXchange with identifier PXD047232.

# Peer review information

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

## Acknowledgements

Work at the Novo Nordisk Foundation Center for Protein Research is supported by grant NNF14CC0001 and JN and MM is supported by a grant from Sygeforsikring Danmark and JN by a grant from Independent Research Fund Denmark (3101-00358B). We thank the protein production and characterization unit at NNF CPR for help with purifying proteins and the medical faculty Umeå University strategic research resource and the Laboratory for Molecular Infection Medicine Sweden for generous support (AKÖ), and the Biochemical Imaging Center at Umeå University and the National Microscopy Infrastructure, NMI (VR-RFI 2016-00968) for assistance in microscopy. Work in AKÖ lab is supported by the Swedish Research Council (2018-05851). Research at UTMB was supported by grants from NIAID of the United States NIH to (R01-AI153602, R21-AI145400, U19-AI171413, R24-AI120942) to VDM. The research was also supported by STARs Award provided by the University of Texas System to VDM. Trainee funding provided by NIAID of the NIH to MNV (T32-AI060549). We thank Anne-Claude Gingras for providing the HeLa UBAP2L KO cell line.

## Author contributions

**Dimitriya H Garvanska**: Conceptualization; Formal analysis; Investigation; Visualization; Methodology; Project administration; Writing—review and editing. **R Elias Alvarado**: Formal analysis; Investigation; Visualization; Methodology; Project administration. **Filip Oskar Mundt**: Formal analysis; Investigation; Visualization; Methodology. **Richard Lindqvist**: Formal analysis; Investigation; Visualization; Methodology. **Josephine Kerzel Duel**: Formal analysis; Investigation; Visualization; Methodology. **Fabian Coscia**: Formal analysis; Investigation; Visualization; Methodology. **Emma Nilsson**: Formal analysis; Investigation; Visualization; Methodology. **Kumari Lokugamage**: Formal analysis; Investigation; Visualization; Methodology. **Bryan A Johnson**: Formal analysis; Investigation; Visualization; Methodology. **Jessica A Plante**: Formal analysis; Investigation; Visualization; Methodology. **Dorothea R Morris**: Formal analysis; Investigation; Visualization; Methodology. **Michelle N Vu**: Formal analysis; Validation; Investigation; Methodology. **Leah K Estes**: Formal analysis; Investigation; Visualization; Methodology. **Alyssa M McLeland**: Formal analysis; Investigation; Visualization; Methodology. **Jordyn Walker**: Formal analysis; Investigation; Visualization; Methodology. **Patricia A Crocquet-Valdes**: Formal analysis; Investigation; Visualization; Methodology. **Blanca Lopez Mendez**: Formal analysis; Investigation; Visualization; Methodology. **Kenneth S Plante**: Formal analysis; Investigation; Visualization; Methodology. **David H Walker**: Formal analysis; Investigation; Visualization; Methodology. **Melanie Bianca Weisser**: Visualization; Methodology. **Anna K Överby**: Funding acquisition; Project administration; Writing—review and editing. **Matthias Mann**: Funding acquisition; Project administration; Writing—review and editing. **Vineet D Menachery**: Funding acquisition; Writing—original draft; Project administration. **Jakob Nilsson**: Conceptualization; Funding acquisition; Investigation; Visualization; Writing—original draft; Project administration.

## Disclosure and competing interests statement

VDM has filed a patent on the reverse genetic system and reporter SARS-CoV-2. The remaining authors declare no competing interests.

# Expanded View Figures

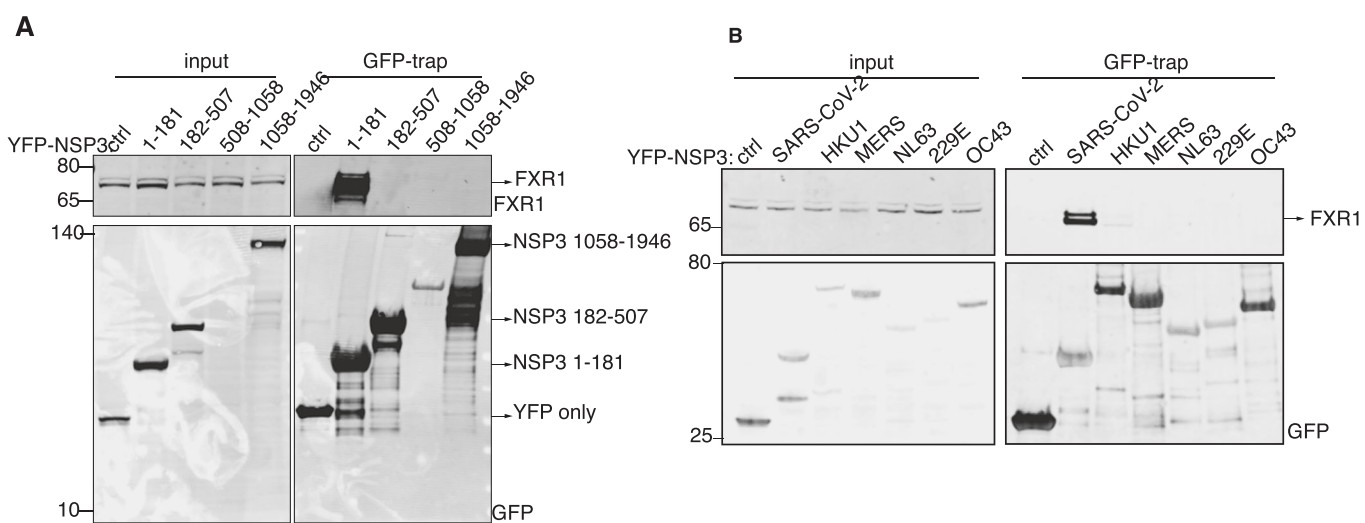

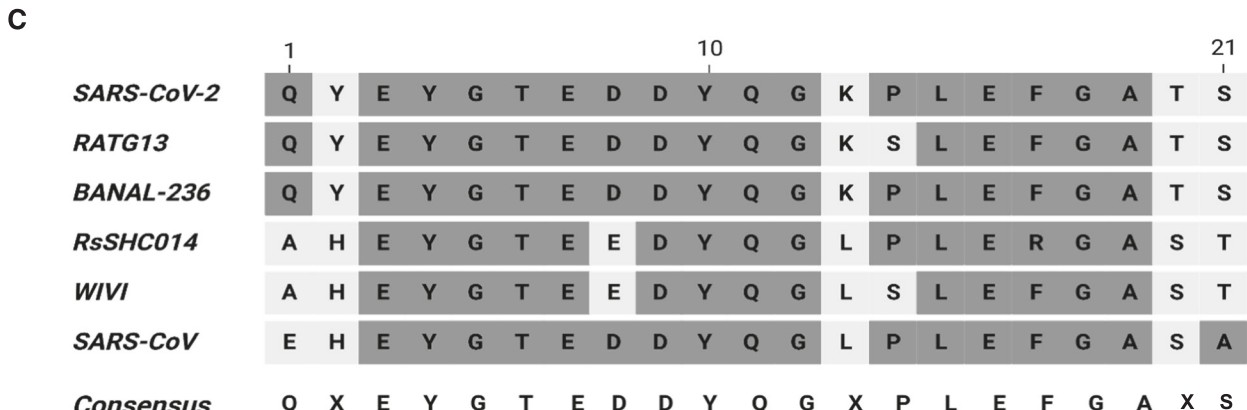

**Figure EV1. Analysis of FMRP-NSP3 interaction.**

(A) The indicated NSP3 fragments fused to YFP was expressed and purified from HeLa cells and binding to FXR1 monitored. Representative of two biological replicates. (B) A panel of NSP3 N-terminal fragments from different coronaviruses were expressed and purified from HeLa cells and binding to FXR1 determined by western blotting. Representative of two biological replicates. (C) Alignment of the NSP3 sequence binding to FMRPs from different coronaviruses.

## A

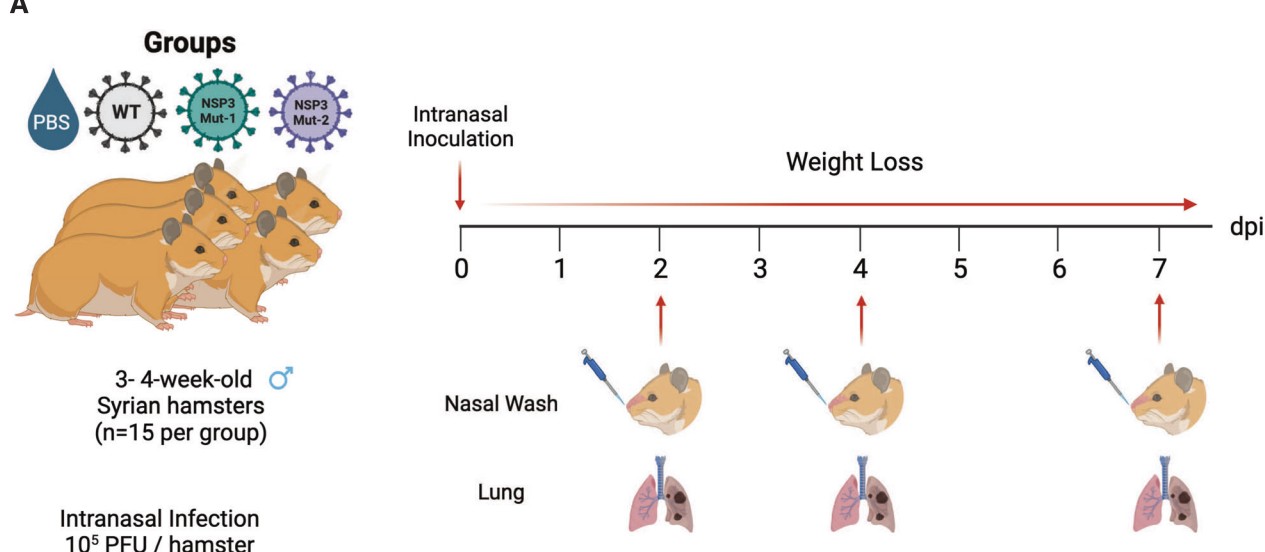

## B

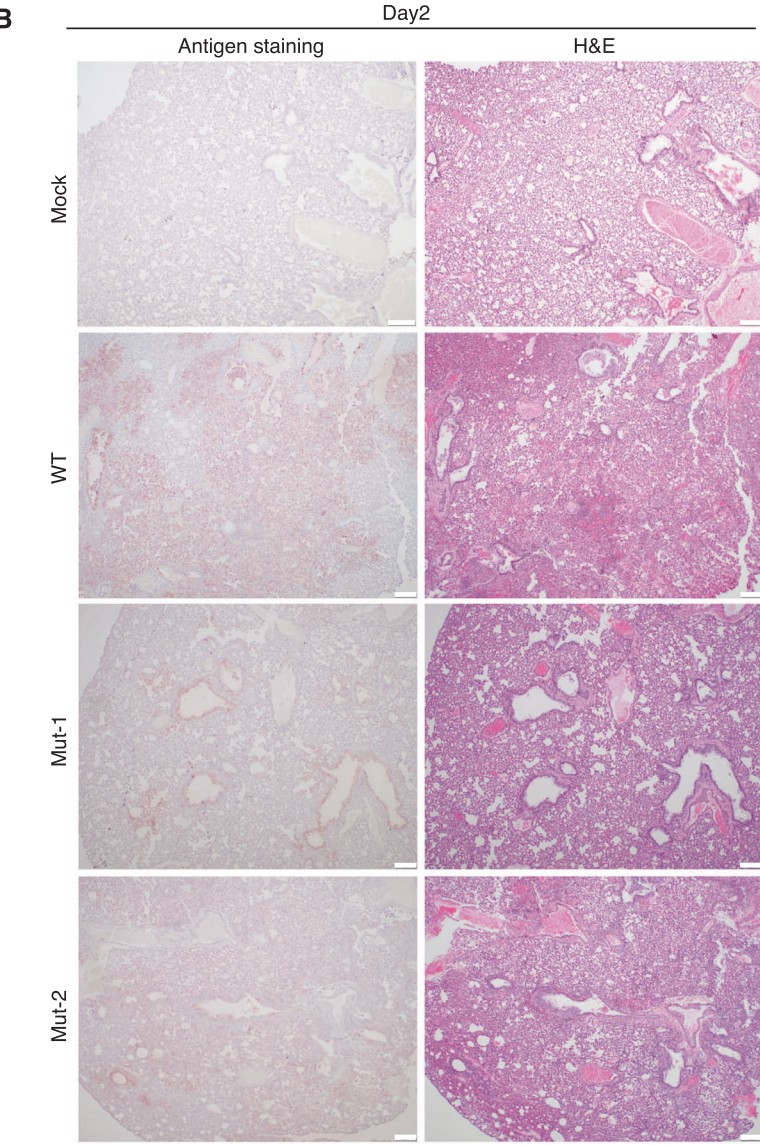

◀ **Figure EV2. Histopathology of hamster infected with WT or NSP3 Mutants.**

(A) Schematic of in vivo experiment (generated with BioRender). (B) H&E and viral antigen (nucleocapsid) immunohistochemical staining of lung of hamsters infected mock (PBS) or with $10^5$ pfu of WT, NSP3 Mut-1, or NSP3 Mut-2 SARS-CoV-2 at 2 days post infection. WT infection shows extensive viral infection and damage; both NSP3 mutants have focal disease and less damage. No damage observed in mock infected samples. Images from representative section from a single hamster in each group (mock, WT, NSP3 Mut-1, & NSP3 mut-2). For larger magnification see Fig. 2. Data information: Scale bar of 200 μm is indicated in the lower right corner.

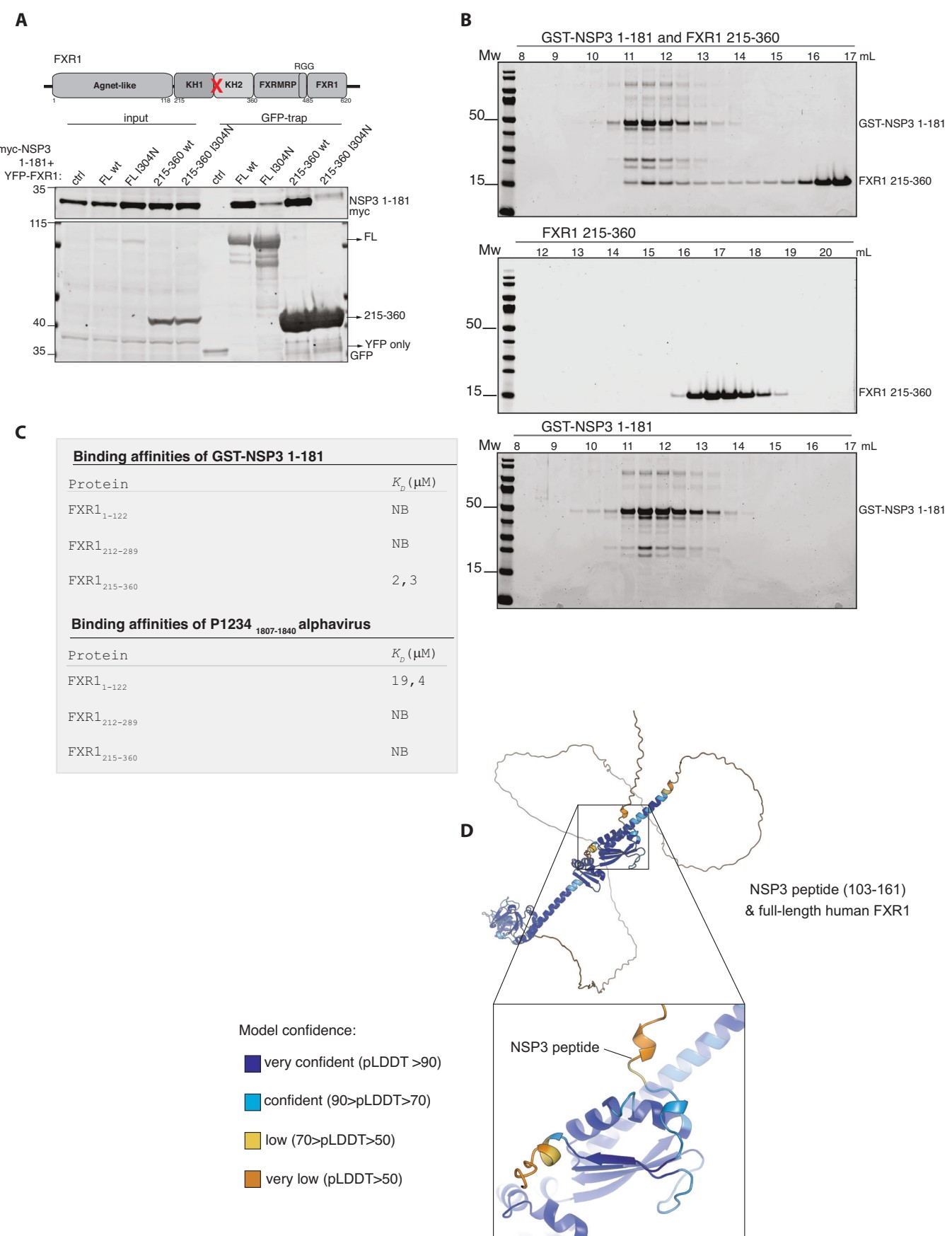

**A**

**C**

**Binding affinities of GST-NSP3 1-181**

| Protein | $K_D$ (µM) |
| --- | --- |
| FXR1$_{1-122}$ | NB |
| FXR1$_{212-289}$ | NB |
| FXR1$_{215-360}$ | 2,3 |

**Binding affinities of P1234 $_{1807-1840}$ alphavirus**

| Protein | $K_D$ (µM) |
| --- | --- |
| FXR1$_{1-122}$ | 19,4 |
| FXR1$_{212-289}$ | NB |
| FXR1$_{215-360}$ | NB |

**B**

GST-NSP3 1-181 and FXR1 215-360

FXR1 215-360

GST-NSP3 1-181

**D**

NSP3 peptide (103-161)
& full-length human FXR1

Model confidence:

very confident (pLDDT >90)

confident (90>pLDDT>70)

low (70>pLDDT>50)

very low (pLDDT>50)

NSP3 peptide

**Figure EV3.   A direct interaction between NSP3 and FXR1.**

(A) The indicated FXR1 YFP constructs were co-expressed with myc-NSP3 1–181 in HeLa cells and affinity-purified using YFP affinity beads. The binding to NSP3 was monitored by probing for myc. Representative of two biological replicates. (B) Size-exclusion chromatography of GST-NSP3 WT 1–181, FXR1 215–360 either alone or in combination. The elution volume is indicated on top and Coomasie stained gels of fractions shown. Representative of two biological replicates. (C) Table of ITC values obtained for the indicated FXR1 fragments binding to GST-NSP3 1–181 or the FXR1 binding peptide from old alphaviruses ($n = 1$). (D) Confidence plots of AlphaFold model of NSP3 peptide binding to full-length FXR1.

**A**

### Binding affinities of RNA

| Protein | $K_D\,(\mu M)$ |
|---|---|
| FXR1$_{215-360}$ | NB |
| N protein | 0,4 |

**B**

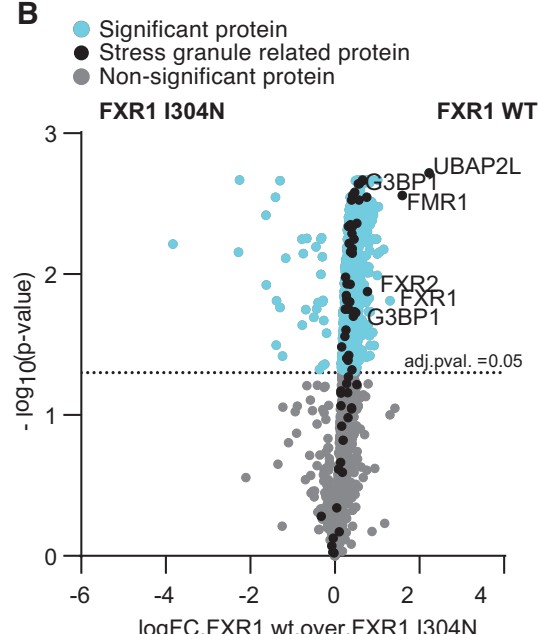

**C**

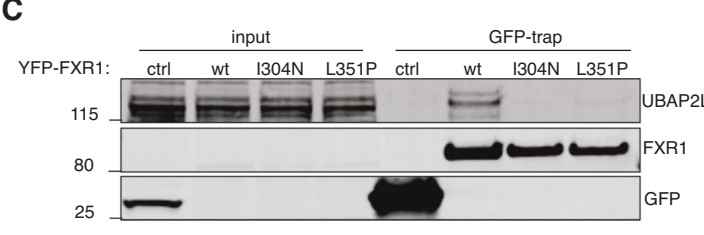

**D**

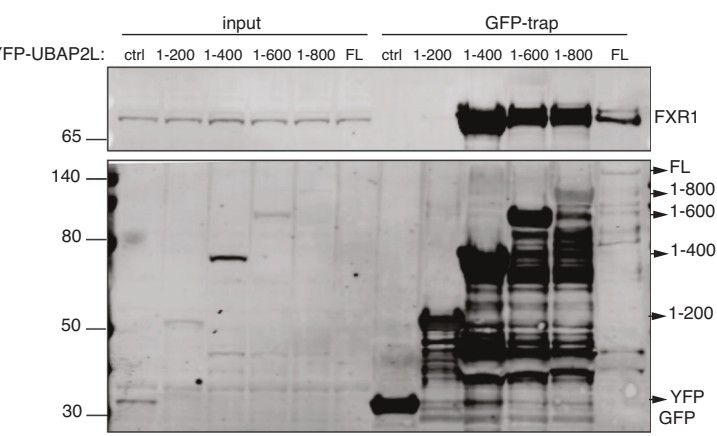

**Figure EV4.   Interaction of FXR1 to UBAP2L.**

(A) ITC measurements of a reported RNA binding to FXR1. Binding to FXR1 215–360 was monitored and as a control the SARS-CoV-2 N protein (n = 1). (B) Mass spectrometry analysis of the interactomes of affinity-purified YFP-tagged FXR1 WT and FXR1 I304N. Proteins specifically binding to FXR1 WT indicated in the volcano plot. Data from 4 technical repeats. (C) The indicated YFP-tagged FXR1 proteins were expressed and purified from HeLa cells and binding to UBAP2L determined by western blot. Representative of 2 biological replicates. (D) A panel of YFP-UBAP2L constructs were expressed and purified from HeLa cells and binding to FXR1 determined. Representative of 2 biological replicates. Data information: In (B) a two-sided unpaired t test was used for statistical analysis.

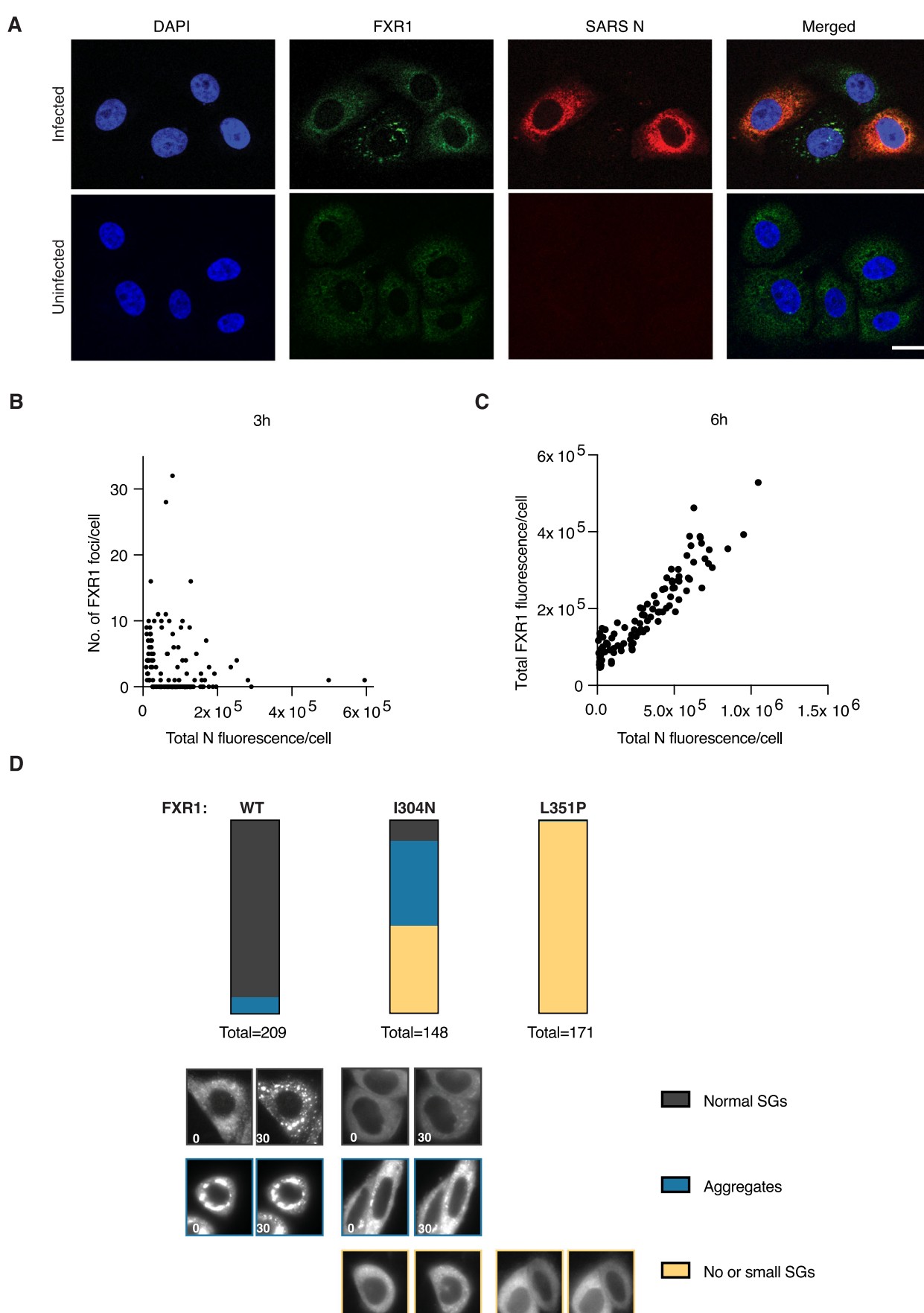

◄ **Figure EV5.  Analysis of FXR1 localization to stress granules.**

(**A**) VeroE6 cells infected with SARS-CoV-2 or uninfected were fixed and stained for FXR1 and the viral N protein. Representative images shown from one experiment. (**B**) The number of FXR1 foci in infected cells was determined and correlated with total level of N protein. (**C**) The total level of FXR1 was determined in infected cells and plotted against total levels of N. (**D**) YFP-tagged FXR1 proteins were expressed in HeLa cells and filmed by live-cell microscopy. Stress granule formation was induced by arsenite and 30 min after addition the localization and morphology of FXR1 foci was monitored. Phenotypes are plotted as percentage. Scores of two individual experiments are shown. The total number of cells analyzed per condition are indicated. Representative images are shown. Data information: In (**A**) scale bar is 20 μm and in (**D**) it is 10 μm.

