## [Peer Review File · EMBO Reports]

The NSP3 protein of SARS-CoV-2 binds fragile X mental retardation proteins to disrupt UBAP2L interactions.

Dimitriya Garvanska, Rojelio Alvarado, Filip Mundt, Richard Lindqvist, Josephine Duel, Fabian Coscia, Emma Nilsson, Kumari Lokugamage, Bryan Johnson, Jessica Plante, Dorothea Morris, Michelle Vu, Leah Estes, Alyssa McLeland, Jordyn Walker, Patricia Crocquet-Valdes, Blanca Lopez-Mendez, Kenneth Plante, David Walker, Melanie Weisser, Anna Överby, Matthias Mann, Vineet Menachery, and Jakob Nilsson

DOI: [10.15252/embr.202358481](https://doi.org/10.15252/embr.202358481)

Corresponding author(s): Jakob Nilsson (jakob.nilsson@cpr.ku.dk)

Review Timeline:

Transfer Date:	13th Nov 23
Editorial Decision:	14th Nov 23
Revision Received:	27th Nov 23
Editorial Decision:	12th Dec 23
Revision Received:	14th Dec 23
Accepted:	14th Dec 23

Editor: Achim Breiling

Transaction Report: This manuscript was transferred to EMBO reports following peer review at The EMBO Journal.

Dear Dr. Nilsson

Thank you for transferring your manuscript to EMBO reports. I now went through your manuscript, the two referee reports from The EMBO Journal (attached again below) and your preliminary point-by-point-response (revision plan). The referees have several concerns and suggestions to improve the manuscript, or to strengthen the data and the conclusions drawn.

Given the constructive referee comments, I would like to invite you to revise your manuscript with the understanding that all concerns of the referees must be addressed in the revised manuscript or in a detailed point-by-point response, as indicated in your revision plan. Acceptance of your manuscript will depend on a positive outcome of another round of review at EMBO reports, using the same referees.

1) a .docx formatted version of the final manuscript text (including legends for main figures, EV figures and tables), but without the figures included. Please make sure that changes are highlighted to be clearly visible. Figure legends should be compiled at the end of the manuscript text.

2) individual production quality figure files as .eps, .tif, .jpg (one file per figure), of main figures and EV figures. Please upload these as separate, individual files upon re-submission. Please make sure that all figure panels are called out separately and sequentially in the manuscript text

For more details please refer to our guide to authors:

See also our guide for figure preparation:

Moreover, please consult our guidelines for figure legend preparation:

4) a complete author checklist, which you can download from our author guidelines

(<https://www.embopress.org/page/journal/14693178/authorguide>). Please insert page numbers in the checklist to indicate where the requested information can be found in the manuscript. The completed author checklist will also be part of the RPF.

5) that primary datasets produced in this study (e.g. RNA-seq, ChIP-seq and array data) are deposited in an appropriate public database. This is now mandatory (like the COI statement). If no primary datasets have been deposited in any database, please state this in this section (e.g. 'No primary datasets have been generated and deposited').

The accession numbers and database should be listed in a formal "Data Availability " section (placed after Materials & Methods) that follows the model below. Please note that the Data Availability Section is restricted to new primary data that are part of this study.

Data availability

8) Regarding data quantification and statistics, please make sure that the number "n" for how many independent experiments were performed, their nature (biological versus technical replicates), the bars and error bars (e.g. SEM, SD) and the test used to calculate p-values is indicated in the respective figure legends (also for potential EV figures and all those in the final Appendix). Please also check that all the p-values are explained in the legend, and that these fit to those shown in the figure. Please provide statistical testing where applicable. Please avoid the phrase 'independent experiment', but clearly state if these were biological or technical replicates. Please also indicate (e.g. with n.s.) if testing was performed, but the differences are not significant. In case n=2, please show the data as separate datapoints without error bars and statistics.

See also:

<http://www.embopress.org/page/journal/14693178/authorguide#statisticalanalysis>

If $n < 5$, please show single datapoints for diagrams.

9) Please add scale bars of similar style and thickness to microscopic images, using clearly visible black or white bars (depending on the background). Please place these in the lower right corner of the images themselves. Please do not write on or near the bars in the image but define the size in the respective figure legend.

10) Please note our reference format:

11) We updated our journal's competing interests policy in January 2022 and request authors to consider both actual and perceived competing interests. Please review the policy <https://www.embopress.org/competing-interests> and add a statement declaring your competing interests. Please name that section 'Disclosure and Competing Interests Statement' and add it after the author contributions section.

12) Please add up to 5 key words to the manuscript text file (below the abstract) and order the manuscript sections like this using these names:

Title page - Abstract - Keywords - Introduction - Results - Discussion - Materials and Methods - Data availability section (DAS) - Acknowledgements - Disclosure and Competing Interests Statement - References - Figure legends - Expanded View Figure legends

13) Please make sure that all the funding information is also entered into the online submission system and is complete and similar to the one in the manuscript text file (in the Acknowledgements).

14) We now use CRediT to specify the contributions of each author in the journal submission system. CRediT replaces the author contribution section. Please use the free text box to provide more detailed descriptions. Thus, please remove the author contributions section from the final manuscript text file. See also guide to authors:

<https://www.embopress.org/page/journal/14693178/authorguide#authorshipguidelines>

15) We would encourage you to use 'Structured Methods', our new Materials and Methods format. According to this format, the Materials and Methods section should include a Reagents and Tools Table (listing key reagents, experimental models, software and relevant equipment and including their sources and relevant identifiers) followed by a Methods and Protocols section in which we encourage the authors to describe their methods using a step-by-step protocol format with bullet points, to facilitate the adoption of the methodologies across labs. More information on how to adhere to this format as well as downloadable templates (.doc or .xls) for the Reagents and Tools Table can be found in our author guidelines (section 'Structured Methods'):

I look forward to seeing a revised version of your manuscript when it is ready. Please let me know if you have questions or comments regarding the revision.

I look forward to seeing a revised form of your manuscript when it is ready.

Kind regards,

Achim

Referee #1:

Garvanska and colleagues present a detailed molecular characterization of the interaction between SARS-CoV-2 nsp3 and members of the fragile X mental retardation protein family, including FXR1, FXR2 and FMR1 (FRMPs). Nps3 has recently been identified as a viral protein that is critical for the formation of replication complex pores. Ectopic expression of YFP-nsp3 followed by purification and mass spectrometry analysis revealed that all three FRMPs are co-purified, confirming previous research. Using sophisticated in vitro and cellular biochemical approaches, the authors delved deeper into the regions responsible for binding and narrow down to a 20aa-stretch in the hypervariable region of nsp3, alongside a conserved region among FRMPs that contain two KH domains. The authors identified two crucial amino acid residues, F145 in nsp3 and I304 FXR1, whose mutation disrupts the interaction between FXR1 and nsp3. They conducted further analysis to investigate whether the binding of nsp3 outcompetes other FXR1 interaction partners. Using mass spectrometry, they observed a decrease in proteins linked to stress granules, such as UBAP2L. I258 and F259 have been identified as critical UBAP2L residues that bind to the FXR1 domain, which is also capable of binding to nsp3. In cells, expression of nsp3 impairs the recruitment of FXR1 into stress granules.

Together, the study is convincing and provides a comprehensive molecular analysis of how nsp3 interacts with FXR1, which impairs its binding to UBAP2L and recruitment into stress granules during stress. The characterization of the critical residues involved in each of the proteins is new and provides a clearer picture of the mechanism. Experiments are technically sound and well controlled. Data are convincing and of good scientific quality. The authors conclude their manuscript suggesting a viral mechanism to counteract the antiviral function of stress granules at early stages of infection. However, this link between stress granules and the immune response in the context of the mutant virus is not investigated, and although this reviewer acknowledges the quality of the scientific approach, the in vivo results show that this interaction is of little importance to the outcome of SARS-CoV-2 infection. Due to the limited in vivo relevance and the lack of evidence linking this interaction to the antiviral function of stress granules, the overall significance is reduced and the study aimed at a more specialized readership than that of the EMBO Journal.

The specific comments are as follows:

- The title of the manuscript is not explicit enough; it is even misleading since the effect on the virus is very modest.
- The authors should mention that the interaction between UBAP2L and FXR1 was previously reported by Sanders et al, 2020 Cell (PMID 32302570).
- Supplementary Figure 6: The authors analyze the localization and levels of FXR1 in SARS-CoV2 infected cells. Cells

expressing low levels of N protein appear negative on the still image, probably because N protein accumulates later, as they mention. Staining the dsRNA with the J2 antibody would avoid this discrepancy and allow this to be analyzed more finely. The panel showing uninfected cells should be added. The authors report an increase in total FXR1 levels, as analyzed by fluorescence microscopy. Is this increase also confirmed by Western blot analysis?

- Figure 5A, control panels for untreated and untransfected cells are missing. Upon expression of YFP-nsp3 1-181, the signal for FXR1 disappears. This contradicts the previous observation that infection increases FXR1 levels. Is this phenomenon also observed in total FXR1 levels?

Referee #2:

General summary and opinion about the principle significance of the study, its questions and findings:

The presented work by Garvanska et al. titled SARS-CoV-2 hijacks fragile X mental retardation proteins for efficient infection identified specific interaction between the cytoplasmic N-terminal region of SARS-CoV-2 non-structural protein 3 (Nsp3, a.k.a. PLpro) and the host fragile X mental retardation proteins (FMR1, FXR1, and FXR2 collectively referred to as FMRPs). They extensively characterize binding sites on Nsp3 and FMRPs using co-immunoprecipitation and peptide binding assays. This work identifies residues in the region of Nsp3 spanning amino acids 129-148 that are essential for FMRP binding. They also convincingly show that Nsp3 competes with UBAP2L protein for binding to the same pocket in FMRPs KH domains that disrupts UBAP2L-mediated recruitment of FMRPs to stress granules (SGs). Notably, mutation in KH domain of FXR1 associated with fragile X mental retardation disease I304N abolishes binding to either UBAP2L or Nsp3, confirming that both interactors use the same binding pocket and suggesting that the loss of FXR1-UBAP2L interaction may be contributing to disease phenotype. Finally, they show that mutant SARS-CoV-2 viruses with alanine substitutions in Nsp3 region that disrupt FMRP binding are modestly attenuated in in vitro cell culture and in vivo hamster infection models.

Specific major concerns essential to be addressed to support the conclusions:

1. The data presented in this study is generally of high quality. Biochemical evidence for Nsp3-FMRP interaction is strong and the studies in cells nicely show disruption of FMRP trafficking to SGs by Nsp3. However, the link between loss of FMRP binding and mild virus attenuation is not convincingly presented. Specifically, it is unclear if or how disruption of FMRP-UBAP2L interaction subverts antiviral SG responses in SARS-CoV-2 infected cells, particularly since FMRP proteins are not essential for SG formation. Given extensive discussion of SGs and their subversion by SARS-CoV-2, the evidence linking SG responses and mutations that disrupt Nsp3-FMRP interaction to virus attenuation is lacking. As presented, the results could be explained by pro-viral role of Nsp3-FMRP interaction rather than antiviral role of SGs. Without additional experimentation, this reviewer suggests modifying discussion and conclusions of this paper to better fit the presented data with less emphasis on the role of SGs.

2. Formally there is a possibility that mutations in the region of amino acids 129-148 of Nsp3 could affect other activities of this protein that are not related to FMRP interaction (e.g. protein folding, stability), causing mild attenuation of mutant viruses. This possibility should be mentioned/discussed in the manuscript and potential experiments addressing the issue proposed (e.g. infection of UBAP2L KO or FMRP KO cell lines, functional assessment of PLpro activity upon overexpression in cells).

Additional non-essential suggestions for improving the study (which will be at the author's/editor's discretion):

Another area where the data discussion should be improved is the in vivo results. Indeed, there appears to be a measurable decrease in viral antigen staining in the lungs. However, the virus titers in lungs and nasal washes are very similar between the wild type and the mutant viruses, and animals have very similar weight loss dynamics. This apparent discrepancy in results is not well described. Pathology scores based on H&E staining do not show the full picture and there appears to be a missed opportunity to characterize immune responses in the lungs in more depth.

Referee #1:

Garvanska and colleagues present a detailed molecular characterization of the interaction between SARS-CoV-2 nsp3 and members of the fragile X mental retardation protein family, including FXR1, FXR2 and FMR1 (FRMPs). Nsp3 has recently been identified as a viral protein that is critical for the formation of replication complex pores. Ectopic expression of YFP-nsp3 followed by purification and mass spectrometry analysis revealed that all three FRMPs are co-purified, confirming previous research. Using sophisticated in vitro and cellular biochemical approaches, the authors delved deeper into the regions responsible for binding and narrow down to a 20aa-stretch in the hypervariable region of nsp3, alongside a conserved region among FRMPs that contain two KH domains. The authors identified two crucial amino acid residues, F145 in nsp3 and I304 FXR1, whose mutation disrupts the interaction between FXR1 and nsp3. They conducted further analysis to investigate whether the binding of nsp3 outcompetes other FXR1 interaction partners. Using mass spectrometry, they observed a decrease in proteins linked to stress granules, such as UBAP2L. I258 and F259 have been identified as critical UBAP2L residues that bind to the FXR1 domain, which is also capable of binding to nsp3. In cells, expression of nsp3 impairs the recruitment of FXR1 into stress granules.

Together, the study is convincing and provides a comprehensive molecular analysis of how nsp3 interacts with FXR1, which impairs its binding to UBAP2L and recruitment into stress granules during stress. The characterization of the critical residues involved in each of the proteins is new and provides a clearer picture of the mechanism. Experiments are technically sound and well controlled. Data are convincing and of good scientific quality. The authors conclude their manuscript suggesting a viral mechanism to counteract the antiviral function of stress granules at early stages of infection. However, this link between stress granules and the immune response in the context of the mutant virus is not investigated, and although this reviewer acknowledges the quality of the scientific approach, the in vivo results show that this interaction is of little importance to the outcome of SARS-CoV-2 infection. Due to the limited in vivo relevance and the lack of evidence linking this interaction to the antiviral function of stress granules, the overall significance is reduced and the study aimed at a more specialized readership than that of the EMBO Journal.

Our response:

We thank the reviewer for the overall positive comments on our work and its quality. Indeed, the effects on SARS-CoV-2 are modest as we also state. This is likely due to redundant functions of the N-G3BP1/2 and NSP3-FMRP interactions in antagonizing stress granule functions. We anticipate that both viral stress granule interaction motifs must be mutated to observe a strong phenotype. However, we suggest that NSP3-FMRP plays an important role during the early parts of infection when N protein is less abundant; as N protein

accumulates, it asserts control over stress granule responses. This would explain the kinetic delay in pathogenesis, but overall similar disease in the NSP3 mutants versus WT.

Our results are similar to what has been observed in alphaviruses but beyond the current work to investigate. We want to point out that very few studies have investigated the effect of mutating host factor binding sites in SARS-CoV-2 and its effect in vivo so it is difficult to compare the effects we see to other studies.

We have in the revised manuscript pointed out very clearly that the effects are not strong and that this might be due to redundant functions of N and NSP3. Furthermore, we point out in the revised discussion that we cannot exclude that the function of the NSP3-FMRP interaction is unrelated to an effect on stress granules and that the function of the complex could be pro-viral.

The specific comments are as follows:

- The title of the manuscript is not explicit enough, it is even misleading since the effect on the virus is very modest.

Our response: We have changed the title to be more explicit.

- The authors should mention that the interaction between UBAP2L and FXR1 was previously reported by Sanders et al, 2020 Cell (PMID 32302570).

Our response: We have included this reference. While we are not the first to observe UBAP2L-FMRP interactions, to our knowledge we are the first to show that this is a direct protein-protein interaction. We also provide amino acid resolution information on the interaction providing potential insight into disease mutations in FMRPs.

- Supplementary Figure 6: The authors analyze the localization and levels of FXR1 in SARS-CoV2 infected cells. Cells expressing low levels of N protein appear negative on the still image, probably because N protein accumulates later, as they mention. Staining the dsRNA with the J2 antibody would avoid this discrepancy and allow this to be analyzed more finely. The panel showing uninfected cells should be added. The authors report an increase in total FXR1 levels, as analyzed by fluorescence microscopy. Is this increase also confirmed by Western blot analysis?

Our response: We have added panels for uninfected cells. We have used N protein expression as an indication of infection stage as described previously (Kruse et al, NCOMMS 2021). We did stain for dsRNA in pilot experiments, but the results were similar to N staining, so we decided to use N staining (see image below).

Infected (top) and uninfected cells (bottom) stained for N protein and dsRNA (viral).

We have not analyzed FXR1 levels by WB but a previous mass spectrometry study that we cite (Stukalov et al, Nature 2021) have observed the same. For this reason, we did not perform the western blot analysis. Below is the data on FXR1 from Stukalov et al:

A figure was shown here, presenting protein expression data for FXR1, taken from the site <https://covinet.innatelab.org>, giving access to the large dataset published in:

Stukalov et al. (2021), Nature 594(7862), Multilevel proteomics reveals host perturbations by SARS-CoV-2 and SARS-CoV, doi: 10.1038/s41586-021-03493-4

- Figure 5A, control panels for untreated and untransfected cells are missing. Upon expression of YFP-nsp3 1-181, the signal for FXR1 disappears. This contradicts the previous observation that infection increases FXR1 levels. Is this phenomenon also observed in total FXR1 levels?

Our response: We think the relevant comparison here is YFP-NSP3 1-181 wild type to mutant as this is the most precise comparison and then comparing effects on FXR1 and G3BP1/2. We do not see FXR1 incorporating into stress granules without adding arsenite (see image below from previous supplemental Fig. 7 that is a time-lapse experiment following stress granule formation after 30 min of addition of arsenite, before addition there is no stress granules) and we have tested that YFP expression does not affect FXR1 incorporation (images now added to Figure 5).

HeLa

Incorporation of G3BP1 and FXR1 depends on addition of arsenite.

We want to point out that the increase in FXR1 levels observed during infection is not observed in these experiments. The FXR1 signal does not disappear in these experiments but simply disperses. We think that this does not contradict the virus results on FXR1 levels as these are very different types of experiments. We have not claimed that the increase in FXR1 levels during infection is related to NSP3-FMRP interaction.

Referee #2:

- general summary and opinion about the principle significance of the study, its questions and findings

The presented work by Garvanska et al. titled SARS-CoV-2 hijacks fragile X mental retardation proteins for efficient infection identified specific interaction between the cytoplasmic N-terminal region of SARS-CoV-2 non-structural protein 3 (Nsp3, a.k.a. PLpro) and the host fragile X mental retardation proteins (FMR1, FXR1, and FXR2 collectively referred to as FMRPs). They extensively characterize binding sites on Nsp3 and FMRPs using co-immunoprecipitation and peptide binding assays. This work identifies residues in the region of Nsp3 spanning amino acids 129-148 that are essential for FMRP binding. They also convincingly show that Nsp3 competes with UBAP2L protein for binding to the same pocket in FMRPs KH domains that disrupts UBAP2L-mediated recruitment of FMRPs to stress granules (SGs). Notably, mutation in KH domain of FXR1 associated with fragile X mental retardation disease I304N abolishes binding to either UBAP2L or Nsp3, confirming that both interactors use the same binding pocket and suggesting that the loss of FXR1-UBAP2L interaction may be contributing to disease phenotype. Finally, they show that mutant SARS-CoV-2 viruses with alanine substitutions in Nsp3 region that disrupt FMRP binding are modestly attenuated in in vitro cell culture and in vivo hamster infection models.

- specific major concerns essential to be addressed to support the conclusions

1. The data presented in this study is generally of high quality. Biochemical evidence for Nsp3-FMRP interaction is strong and the studies in cells nicely show disruption of FMRP trafficking to SGs by Nsp3. However, the link between loss of FMRP binding and mild virus attenuation is not convincingly presented. Specifically, it is unclear if or how disruption of FMRP-UBAP2L interaction subverts antiviral SG responses in SARS-CoV-2 infected cells, particularly since FMRP proteins are not essential for SG formation. Given extensive discussion of SGs and their subversion by SARS-CoV-2, the evidence linking SG responses and mutations that disrupt Nsp3-FMRP interaction to virus attenuation is lacking. As presented, the results could be explained by pro-viral role of Nsp3-FMRP interaction rather than antiviral role of SGs. Without additional experimentation, this reviewer suggests modifying discussion and conclusions of this paper to better fit the presented data with less emphasis on the role of SGs.

Our response: We agree with the reviewer that we cannot directly link effects of NSP3 mutations in the engineered viruses to effect on stress granules. A complicating factor is that the mechanistic role of FMRPs in stress granule responses is not clear and may be cell/tissue type dependent. We see a correlation between NSP3 mutants affecting viral replication in vitro and infection phenotype and that expression of NSP3 affects incorporation of FMRPs into stress granules. However, we at present we can only propose that these observations are directly linked. Therefore, we have in the revised manuscript updated the discussion to state this and provide alternative explanations for the function of the interaction (text from revised discussion below):

“The KH domain of FMRPs are known to bind specific RNAs and disruption of its stress granule incorporation may prevent sequestration of RNAs key to supporting viral replication. At present we cannot exclude that the interaction between NSP3 and FMRPs is unrelated to an effect on stress granules and that it has a pro-viral function that needs to be elucidated. As an example FMRPs may be hijacked by SARS-CoV-2 NSP3 to play a role in viral transcription or translation.”

2. Formally there is a possibility that mutations in the region of amino acids 129-148 of Nsp3 could affect other activities of this protein that are not related to FMRP interaction (e.g. protein folding, stability), causing mild attenuation of mutant viruses. This possibility should be mentioned/discussed in the manuscript and potential experiments addressing the issue proposed (e.g. infection of UBAP2L KO or FMRP KO cell lines, functional assessment of PLpro activity upon overexpression in cells).

Our response: We do not agree with the reviewer on this. Firstly, we see that the binding to N protein is not affected in our NSP3 mutants (Fig. 1C). As N interacts with the globular domain of NSP3 this argues that the folding of this domain is not affected (we have now made this point more clear in the revised manuscript). Secondly, we have purified recombinant

NSP3 WT 1-181 and NSP3 mut1 1-181 and they both migrate at a similar size on size exclusion chromatography distinct from the void volume. Thirdly, the region we are mutating in NSP3 is unstructured and thus mutations in this region are unlikely to cause misfolding. Consistent with this a 20 mer peptide from this region binds FXR1 directly. Fourthly, we test two distinct mutant viruses that both give the same result further supporting the notion that it is binding to FMRPs that is specifically disrupted in these viruses. Finally, this region is highly variable across the coronavirus family, and the FMRP binding motif is only present in Sarbecovirus family. Lack of conservation across the CoV NSP3 suggest minimal impact on the function of other NSP3 domains. Importantly, the revised manuscript offers a plausible role of FMRP playing a pro-viral role as an alternative to modulating stress granule responses, both of which will require further studies.

- any additional non-essential suggestions for improving the study (which will be at the author's/editor's discretion)

Another area where the data discussion should be improved is the in vivo results. Indeed, there appears to be a measurable decrease in viral antigen staining in the lungs. However, the virus titers in lungs and nasal washes are very similar between the wild type and the mutant viruses, and animals have very similar weight loss dynamics. This apparent discrepancy in results is not well described. Pathology scores based on H&E staining do not show the full picture and there appears to be a missed opportunity to characterize immune responses in the lungs in more depth.

Our response: We have expanded on the description of the in vivo experiments in the main body text, the figure legend, and the materials and methods.

Dear Dr. Nilsson,

Thank you for the submission of your revised manuscript to our editorial offices. I have now received the reports from the two referees that I asked to re-evaluate your study, you will find below. As you will see, both referees support the publication of the study in EMBO reports. Both have comments and suggestions to improve the manuscript, I ask you to address in a final revised manuscript.

During our standard image analysis, we detected potential aberrations in the figure set, and we would like to clarify these issues. It seems the image shown in Fig. 2H (WT H&E) is partly identical to (a subset of) image EV2B (WT H&E).

Please check and comment on the reuse of the image. If purposeful re-use of the image has occurred, please state this clearly in the respective figure legends. If you make changes to the figure set, we require a further response describing what you have changed and why.

Moreover, I have these editorial requests I ask you to address in a final revised manuscript:

- I would suggest a slightly modified title to avoid the double use of bind (it is fine that this is a bit longer than 100 characters):
The NSP3 protein of SARS-CoV-2 binds fragile X mental retardation proteins to disrupt UBAP2L interactions

- We now use CRediT to specify the contributions of each author in the journal submission system. CRediT replaces the author contribution section. Please use the free text box to provide more detailed descriptions and do NOT provide your final manuscript text file with an author contributions section. See also our guide to authors:
<https://www.embopress.org/page/journal/14693178/authorguide#authorshipguidelines>

- Please remove the referee access information from the Data Availability section, provide a direct link to the dataset and make sure this is public latest at the day of online publication of the manuscript.

- Please add scale bars of similar style and thickness to all the microscopic images, using clearly visible black or white bars (depending on the background). Please place these in the lower right corner of the images themselves. Please do not write on or near the bars in the image but define the size in the respective figure legend. Presently, there are still some scale bars with text nearby.

- Please make sure that the number "n" for how many independent experiments were performed, their nature (biological versus technical replicates), the bars and error bars (e.g. SEM, SD) and the test used to calculate p-values is indicated in the respective figure legends (for main, EV and Appendix figures) of the final revised manuscript. Please also check that all the p-values are explained in the legend, and that these fit to those shown in the figure. Please provide statistical testing where applicable. Please avoid the phrase 'independent experiment', but clearly state if these were biological or technical replicates. Please also indicate (e.g. with n.s.) if testing was performed, but the differences are not significant. In case n=2, please show the data as separate datapoints without error bars and statistics. See also:

<http://www.embopress.org/page/journal/14693178/authorguide#statisticalanalysis>

If n<5, please show single datapoints for diagrams. Could statistical testing also be done for panels 1F (please also explain 5.7x, 12.2x and 18.6x shown in the diagram in the legend), 2A, 2B and 2C.

- You state at least twice: 'n=6 from 2 experiments with 3 biological replicates'. Are these then 6 biological replicates? Or do you mean 2 biological replicates with 3 technical repeats? Please clarify.

Moreover:

- Please indicate the statistical test used for data analysis in the legends of figures 1b; 4a-b; EV 4b.

- Please defined the box plots in terms of minima, maxima, centre, bounds of box and whiskers, and percentile in the legends of figures 2e-g.

- Please define the error bars in the legends of figures 1d-f; 2a-c.

- Please correct the scale bar unit in figures 2d, 2h and EV 2b from μM to μm (both in the figure legend and the figure file).

- Please correct the scale bar unit in figures 5a-d; EV 5a, d from μM to μm .

- Please define the yellow arrows in the legend of figure EV 3b.

- Please format the figure legends for main, EV and Appendix figures) according to our journal style. See the respective section in our guide to authors (please find the link below). Please separate each panel description by a line brake and make sure that the panels are listed in alphabetic order. Moreover, please add to each legend a 'Data Information' section explaining the statistics used or providing information regarding replicates and scales.

- Please add page numbers to the Appendix pdf, and a table of contents with page numbers. Please also name the Appendix items Appendix Figure Sx or Appendix Table Sx and adjust their callouts.
- Please move the primer information from the methods section as a table to the Appendix (Appendix Table S1). Please add this to the Appendix TOC, add a title and a legend. Finally, please add a callout for this table to the methods section.
- There are 2 Datasets uploaded. Please name the file for dataset 1 as Dataset EV1. Please add a title and a legend on the first TAB of the excel file. Finally, please also correct the callout for this dataset (Dataset EV1).
- Please move the second dataset to the Appendix and list it there as Appendix table, using the name Appendix Table S2. Please add this to the Appendix TOC, add a title and a legend. Finally, please change the callouts in the manuscript text file.
- Please make sure that all figure panels are called out separately and sequentially (main, EV and Appendix figures). Presently, there seems to be no callout for Fig. 5B. Please check.
- Please add a paragraph titled 'Biosafety' to the methods section gathering all information on where and how biosafety-relevant experiments with viruses were performed and that these were approved, and by whom (institution, government).
- Please make sure that all the funding information is also entered into the online submission system and that it is complete and similar to the one in the acknowledgement section of the manuscript text file. Presently, Sygeforsikring Danmark, Independent Research Fund Denmark, Swedish Research Council (2018-05851), AI171413, STARs Award provided by the University of Texas System are only mentioned in the acknowledgements. Please check.
- Please upload the source data separated, one folder per figure.

In addition, I would need from you:

Best,

Referee #1 (Referee #2 TEJ):

1. The data presented in this study is generally of high quality. Biochemical evidence for Nsp3-FMRP interaction is strong and the studies in cells nicely show disruption of FMRP trafficking to SGs by Nsp3. This reviewer's concern from a previous round of review that the link between loss of FMRP binding and mild virus attenuation could be due to pro-viral role of FMRP has been addressed adequately by including the corresponding clause in the discussion.
2. Despite providing reasonably convincing rebuttal to other concerns raised in the previous round of reviews, the issue of multiple effects of mutations in a key viral protein on viral fitness not directly linked to FMRP interaction is still possible. These FMRP-independent effects formally may, in opinion of this reviewer, be partially or fully responsible for observed mild attenuation of mutant viruses. Infection experiments using FMRP or UBAP2L KO cell line would convincingly alleviate these concerns, but may be beyond the scope of the present work. For the manuscript to be accepted, inclusion of the statement acknowledging this limitation and proposing such experiments in future studies is needed.

Referee #2 (Referee #1 TEJ):

Garvanska and colleagues have addressed most of my previous concerns, particularly by incorporating control panels, revising some of the previously unsubstantiated conclusions regarding the connection between the antiviral function of stress granules and changing the title of their manuscript accordingly. However, although the authors mention that they have included it, I could not find the reference Sanders et al, 2020 Cell (PMID 32302570). Whilst I do agree that the authors "provide information on the amino acid resolution of the interaction, leading to a better understanding of pathological mutations in FMRPs", I consider that the demonstration by others of an interaction between UBAP2L and FXR1 or FMR1 by co-immunoprecipitation experiments under conditions including RNase treatment (see Figures 2D and 2E in Sanders et al) should be accepted as evidence for direct protein-protein interaction. In addition, this paper by Sanders et al. also provided evidence for the formation of distinct UBAP2L protein complexes with G3BP1 or FRMPs that should be discussed. Mentioning this work in no way detracts from the interest of this present study.

Referee #1 (Referee #2 TEJ):

1. The data presented in this study is generally of high quality. Biochemical evidence for Nsp3-FMRP interaction is strong and the studies in cells nicely show disruption of FMRP trafficking to SGs by Nsp3. This reviewer's concern from a previous round of review that the link between loss of FMRP binding and mild virus attenuation could be due to pro-viral role of FMRP has been addressed adequately by including the corresponding clause in the discussion.

We thank the reviewer for acknowledging this.

2. Despite providing reasonably convincing rebuttal to other concerns raised in the previous round of reviews, the issue of multiple effects of mutations in a key viral protein on viral fitness not directly linked to FMRP interaction is still possible. These FMRP-independent effects formally may, in opinion of this reviewer, be partially or fully responsible for observed mild attenuation of mutant viruses. Infection experiments using FMRP or UBAP2L KO cell line would convincingly alleviate these concerns, but may be beyond the scope of the present work. For the manuscript to be accepted, inclusion of the statement acknowledging this limitation and proposing such experiments in future studies is needed.

We agree with the reviewer on this but unfortunately KO cell lines suitable for SARS-CoV-2 infection studies are not available. We have now in the revised discussion written (the added line indicated in bold):

*“At present we cannot exclude that the interaction between NSP3 and FMRPs is unrelated to an effect on stress granules and that it has a pro-viral function that needs to be elucidated. As an example FMRPs may be hijacked by SARS-CoV-2 NSP3 to play a role in viral transcription or translation. **To discriminate between a role in stress granule biology or a pro-viral function it will be critical to conduct infection assays in FMRP or UBAP2L knockout cell lines in future experiments.**”*

Referee #2 (Referee #1 TEJ):

Garvanska and colleagues have addressed most of my previous concerns, particularly by incorporating control panels, revising some of the previously unsubstantiated conclusions regarding the connection between the antiviral

function of stress granules and changing the title of their manuscript accordingly. However, although the authors mention that they have included it, I could not find the reference Sanders et al, 2020 Cell (PMID 32302570). Whilst I do agree that the authors "provide information on the amino acid resolution of the interaction, leading to a better understanding of pathological mutations in FMRPs", I consider that the demonstration by others of an interaction between UBAP2L and FXR1 or FMR1 by co-immunoprecipitation experiments under conditions including RNase treatment (see Figures 2D and 2E in Sanders et al) should be accepted as evidence for direct protein-protein interaction. In addition, this paper by Sanders et al. also provided evidence for the formation of distinct UBAP2L protein complexes with G3BP1 or FRMPs that should be discussed. Mentioning this work in no way detracts from the interest of this present study.

We thank the reviewer for these positive comments – we had referenced Sanders et al in the introduction in the revised manuscript. We disagree that co-immunoprecipitation even in the presence of RNase can be used to support direct protein-protein interactions. One would need to purify the proteins and show that they bind as we do here for the first time. We referenced the Sakai 2011 paper that in our view provided the first indication for a direct interaction and also mapped this to the KH domains. The line stated:

UBAP2L, one of the most affected proteins, shows a strong reduction in co-purification consistent with a possible direct interaction between UBAP2L and FXR1 215-360 shown by a prior two-hybrid screen (Sakai et al, 2011)

We have however now included the Sanders et al reference in the line and added a few additional words to satisfy the reviewer:

UBAP2L, one of the most affected proteins, shows a strong reduction in co-purification consistent with a possible direct interaction between UBAP2L and FXR1 215-360 shown by a prior two-hybrid screen (Sakai et al, 2011) and suggested by the RNase resistant interaction observed in immunopurifications (Sanders et al, 2020)

Jakob Nilsson
University of Copenhagen
The Novo Nordisk Foundation Center for Protein Research
Blegdamsvej 3b, Buld 6
Copenhagen DK-2200
Denmark

Dear Dr. Nilsson,

I am very pleased to accept your manuscript for publication in the next available issue of EMBO reports. Thank you for your contribution to our journal.

Yours sincerely,
